# Functional brain activity constrained by structural connectivity reveals cohort-specific features for serum neurofilament light chain

Saurabh Sihag[1,2], Sébastien Naze [2,3], Foad Taghdiri[4], Melisa Gumus [4], Charles Tator[5,6], Robin Green[5,7], Brenda Colella[7], Kaj Blennow [8,9], Henrik Zetterberg [8,9,10,11], Luis Garcia Dominguez[12,13], Richard Wennberg [5,12], David J. Mikulis[5,14], Maria C. Tartaglia [4,5,12] & James R. Kozloski [2✉]

## Abstract

**Background** Neuro-axonal brain damage releases neurofilament light chain (NfL) proteins, which enter the blood. Serum NfL has recently emerged as a promising biomarker for grading axonal damage, monitoring treatment responses, and prognosis in neurological diseases. Importantly, serum NfL levels also increase with aging, and the interpretation of serum NfL levels in neurological diseases is incomplete due to lack of a reliable model for age-related variation in serum NfL levels in healthy subjects.

**Methods** Graph signal processing (GSP) provides analytical tools, such as graph Fourier transform (GFT), to produce measures from functional dynamics of brain activity constrained by white matter anatomy. Here, we leveraged a set of features using GFT that quantified the coupling between blood oxygen level dependent signals and structural connectome to investigate their associations with serum NfL levels collected from healthy subjects and former athletes with history of concussions.

**Results** Here we show that GSP feature from isthmus cingulate in the right hemisphere (r-iCg) is strongly linked with serum NfL in healthy controls. In contrast, GSP features from temporal lobe and lingual areas in the left hemisphere and posterior cingulate in the right hemisphere are the most associated with serum NfL in former athletes. Additional analysis reveals that the GSP feature from r-iCg is associated with behavioral and structural measures that predict aggressive behavior in healthy controls and former athletes.

**Conclusions** Our results suggest that GSP-derived brain features may be included in models of baseline variance when evaluating NfL as a biomarker of neurological diseases and studying their impact on personality traits.

### Plain language summary

Neurofilament light chain (NfL) is a marker released into the blood as a result of central nervous system damage or neurodegeneration. However, we know little about how NfL levels relate to brain structure and activity. Here, we use imaging data and advanced statistical methods to look at the relationship between brain activity and structure in healthy people and former athletes with a history of multiple concussions, and determine whether these can predict NfL levels in the blood. We find the relationship between brain activity and structure and NfL levels is different between the two groups. Our findings help us to understand how brain injury might impact NfL levels and their relationship with brain activity, and could guide how NfL and imaging data are used as tools in research and in the clinic.

[1] University of Pennsylvania, Philadelphia, PA, USA. [2] T.J. Watson IBM Research Center, Health Care and Life Sciences, Yorktown Heights, NY, USA. [3] IBM Research Australia, Melbourne, VIC, Australia. [4] Tanz Centre for Research in Neurodegenerative Diseases, University of Toronto, Toronto, ON, Canada. [5] Canadian Concussion Centre, University Health Network, Toronto, ON, Canada. [6] Division of Neurosurgery, University Health Network, Toronto, ON, Canada. [7] Department of Rehabilitation Sciences, University of Toronto, Toronto, ON, Canada. [8] Institute of Neuroscience and Physiology, Department of Psychiatry and Neurochemistry, The Sahlgrenska Academy at the University of Gothenburg, Mölndal, Sweden. [9] Clinical Neurochemistry Laboratory, Sahlgrenska University Hospital, Mölndal, Sweden. [10] Department of Neurodegenerative Disease, UCL Institute of Neurology, Queen Square, London, UK. [11] UK Dementia Research Institute at UCL, University College London, London, UK. [12] Division of Neurology, University Health Network, Toronto, ON, Canada. [13] Brain and Behaviour Program, The Hospital for Sick Children, Toronto, ON, Canada. [14] Division of Neuroradiology, University Health Network, Toronto, ON, Canada. ✉email: kozloski@us.ibm.com

Neurofilaments are cytoskeleton proteins of neurons and are predominantly found in myelinated axons. NfL is one of three subunits of neurofilament proteins that are released into the cerebrospinal fluid (CSF) and eventually the blood in significant quantities following axonal damage or neurodegeneration[1–4]. Recent advances in immunoassay technologies have enabled reliable detection of NfL in blood and have been utilized in multiple studies to demonstrate high correlation between NfL levels in CSF and blood[5]. An increased concentration of NfL in blood (serum NfL level) or CSF has been reported in numerous studies of neurodegenerative diseases[6] as well as concussion[7,8]. Since collecting blood-based biomarkers is more practical and desirable for extensive clinical trials as compared to CSF-based biomarkers, numerous studies have analyzed serum NfL levels in the context of different neurological disorders such as multiple sclerosis[9,10], dementia[11], progressive supranuclear palsy[12], traumatic brain injury (TBI)[13], Parkinson's disease[14], Alzheimer's disease[15], and Huntington's disease[16]. Changes in NfL levels have also been linked to aging[9,17] and regional atrophy in cortical brain areas[15,18] and are therefore relatable to brain atrophy in aging among people without a recognizable neurological disease. While existing studies provide convincing evidence that serum NfL level is a promising biomarker to detect neurodegeneration in a broad range of clinical applications, the interpretation of these results has been limited by focusing analysis only on detecting the abnormal increase in serum NfL levels and by a lack of studies aimed at identifying the relevant underlying features associated with serum NfL levels[6,19]. For instance, the fundamental mechanism that links aging with serum NfL levels, even among healthy subjects, is unknown[6]. Moreover, while serum NfL levels have been studied in controlled groups in the context of various neurological disorders, insight into the relationship between serum NfL levels and the onset of common neurological symptoms in a healthy group is still lacking[19]. Therefore, it is relevant to search for underlying features that can help ongoing neurological studies model the variance in serum NfL levels among healthy controls and thereby provide a better understanding of both baseline serum NfL levels and how abnormal variation in these levels is mechanistically linked to neurodegenerative disease through these features.

The analysis of brain imaging data has been extensively utilized in neuroscience for independent or joint studies of aging, cognition, and neurological disorders[20–22]. The application of graph signal processing (GSP) tools in neuroscience has recently gained traction because they provide an analytical framework for subject-specific decomposition of functional signals, wherein different components are associated with varying degrees of conformity to the subject's own brain anatomical network[23–25]. The components of the BOLD signal extracted from task-based functional magnetic resonance imaging (fMRI) that are less aligned with, or 'liberal' with respect to, the underlying white matter architecture have been linked with cognitive flexibility[24]. Furthermore, GSP tools have been used to find discriminating features from resting state fMRI and diffusion MRI (dMRI) in autism spectrum disorder[26] and traumatic brain injury[27]. GSP tools have also been used to evaluate the extent of structure-function decoupling for different brain regions[28]. Recent work has also applied GSP tools for the statistical analysis of functional activity with functional connectivity as the underlying graph, thereby implementing a unimodal analysis[29].

Our primary aim was to use statistical analyses to explore whether the associated features extracted from structural and functional brain imaging data are relevant in characterizing the serum NfL levels of healthy controls (HC) vs. former contact sports athletes with a prior history of concussions (ExPro). For both cohorts of subjects, we analyzed energy distributions of resting state fMRI after graph-informed filtering based on white matter connectivity extracted from dMRI. We hypothesized that measures of conformity of the BOLD activity with the underlying white matter anatomy in specific brain areas might reveal associations with serum NfL levels. GSP analytic tools leverage the spectral properties of the graph that represents the white matter anatomy to disentangle BOLD activity into components that either conform to or deviate from it. Low graph frequency components of the BOLD signal associated with a brain area characterize strong alignment between the functional coupling of this area to its underlying anatomical connectivity[24]. In contrast, high graph frequency components of the BOLD signal energy for an area signifies less intermodal alignment, i.e., loss or deviation of the functional coupling of a brain region to its anatomical connectivity[23]. The discriminating high and low graph frequency features between the two groups have been previously studied in Sihag et al.[30], and our focus here was on exploring their relationships with serum NfL levels in HC and ExPro cohorts.

To explore the clinical and neurological interpretation of the GSP features associated with serum NfL in our experiments, we also tested their associations with cognitive scores and structural measures. The two cohorts differed significantly in terms of aggression and depression related personality assessment scores (discussed in Section II.A.1). Therefore, it was of interest to explore the associations of GSP features with amygdala, since this region is instrumental in a broader neural circuit responsible for modulating aggression[31,32] and has been implicated in depression related disorders[33–35]. Furthermore, we hypothesized that the links between white matter degeneration and serum NfL levels might also be characterized by reduced cortical thickness[36]. In this context, we hypothesized that those GSP features aligned to our hypothesis of serum NfL being associated with conformity of BOLD activity to white matter anatomy might also be associated with cortical thickness measures. In a broader context, cortical thinning is also neurologically and clinically relevant, as it has been associated with structural abnormalities after TBI[37] as well as pathological personality traits[38].

Our results show that both low and high graph frequency components from different brain areas are relevant as features for the prediction of serum NfL levels in both HC and ExPro groups. Even more importantly, GSP features have *different associations* with serum NfL levels across the two groups and thus the statistical models for serum NfL levels are *group specific*: their performances do not broadly generalize to the combined dataset of HC and ExPro subjects. This observation is corroborated by our experiments on the behavioral scores under the umbrella of Personality Assessment Inventory (PAI) and structural metrics for both cohorts. The most striking observation is observed for the region of the isthmus cingulate in the right hemisphere, which shows distinct behavior in predicting NfL and associations with cognitive measures and structural measures in the two cohorts. Specifically, the GSP feature from this region is significantly associated with serum NfL levels in HC subjects but not in ExPro subjects. Moreover, our experiments indicate that this GSP feature has a suppressing statistical effect on the relationship between age and PAI Aggression score only among HC subjects. In ExPro subjects, this GSP feature has significant correlations with the volume of right amygdala (which is observed to be moderated by serum NfL levels in this cohort) and thickness of pericalcarine area in the right hemisphere (which is observed to be negatively correlated with serum NfL only in HC cohort). The two structural measures, volume of right amygdala and thickness of pericalcarine cortex, are associated with aggression as measured by PAI in both healthy subjects and pathological contexts[31,39,40], and therefore, our findings imply

**Table 1 PAI Depression and Aggression subscale scores for HC and ExPro cohorts.**

| PAI scale | Score (HC) | Score (ExPro) | Rank sum statistic | p value (FDR corrected) |
|---|---|---|---|---|
| Depression | 9.7 ± 10 | 17.81 ± 13 | 1218 | 0.0059 |
| Aggression | 9.2 ± 4.93 | 15.64 ± 9.54 | 1204.5 | 0.0234 |

Wilcoxon rank sum test (p value < 0.05 after FDR correction for multiple comparisons).

some significance to the GSP feature from isthmus cingulate in the right hemisphere in understanding behavior.

## Methods

**Participants.** The study was approved by the research ethics boards of the University Health Network (IRB approval reference number: IRB 11-0088). Written consent was obtained from all subjects before participating in the study. The male healthy control subjects (number = 20, mean age = 49.38 years, standard deviation = 10.94 years) were recruited from the community. The subjects had no history of neurological disorders (e.g., seizure disorder), systemic illnesses known to affect the brain (e.g., diabetes and lupus), psychotic disorder, or known developmental disorders (e.g., attention deficit disorder, dyslexia) nor any lesions appearing on MRI. The male former athletes (number = 36, mean age = 50.64 years, standard deviation = 11.36 years) were former professional football, hockey or boxing athletes with history of multiple concussions (mean = 4.14, standard deviation = 1.7). There was no significant difference between the ages or serum NfL levels of the two groups (Mann Whitney U tests at 0.05 significance level). One subject each from the HC and ExPro group were excluded from the study since their serum NfL level was more than three deviations from the mean serum NfL level in their respective groups. There was no significant difference between the years of education for the two groups (HC: mean number of years of education = 16.4 years, standard deviation = 1.81 years, ExPro: mean number of years of education = 15.82 years, standard deviation = 1.68 years). Furthermore, no significant difference was observed in the cognitive scores in the contexts of memory, language, and visuospatial function for the two cohorts. Differences on inhibitory control, which is an executive function, have been reported previously on this sample[41].

*PAI Assessments.* The Personality Assessment Inventory (PAI) is a widely used and well-validated tool to study personality and psychopathology in brain injury[42]. It assesses Axis I and II disorders, including personality disorder, depression, aggression and anxiety, and includes indices of validity, such as positive and negative impression management. PAI scores on different subscales were evaluated for all participants. We observed statistically significant differences (p-value < 0.05 after false discovery rate (FDR) correction for multiple comparisons) in both raw and T-normalized PAI scores of the two cohorts for the subscales of somatic concerns, Depression, Schizophrenia and Aggression. Among these subscales, the Aggression and Depression subscales are considered valid in the context of TBI, as these were known to be not confounded by transdiagnostic measures characteristic of both psychopathology and neuropathology[42]. Table 1 summarizes the raw scores for HC and ExPro cohorts on the two subscales and the statistics for the Wilcoxon rank sum test for statistical differences between them.

*Diffusion magnetic resonance imaging acquisition and processing.* All structural and resting state scans were performed on a 3 Tesla MRI Scanner (GE Signa HDx, Milwaukee, WI, USA) with a standard 8-channel head coil. A high resolution T1-weighted images were obtained using inversion recovery fast spoiled gradient echo (IR-FSPGR), with the following parameters: 180 slices with 1 mm thickness; 3 ms echo time (TE); 7.8 ms repetition time (TR); 450 ms inversion time (TI); 15 flip angle; 25.6 cm field of view (FOV); 256 × 256 matrix size; $1 \times 1 \times 1$ mm$^3$ voxel size. At least one DWI scan was obtained with diffusion gradients applied across 60 spatial directions ($b = 1000$ s/mm$^2$) as well as 10 non-diffusion weighted (B0) scans. The DWI had the following parameters: 2.4 mm thick axial slices, TR = 14,000 ms, FOV = 23 cm, $2.4 \times 2.4$ mm$^2$ in-plane resolution.

Diffusion MRI data were processed using the SCRIPTS pipeline with parameters as described therein[43]. Pre-processing involved correction for eddy-currents and head motions artifacts using FSL. After alignment of the co-registered dMRI to the T1 image, fiber tracking was performed using the MRtrix3 package. Fiber orientation estimation was performed using Constrained Spherical Deconvolution, and tracks were seeded from the white-gray matter interface. A propagation mask was applied through Anatomically Constrained Tractography (ACT) and streamlines were generated using a probabilistic algorithm using second-order integration over fiber orientation distributions (iFOD2) from 10 million seeds (step size 0.5 mm, maximum curvature 45, length 5–250 mm, FOD amplitude threshold 0.1). Streamlines were then selected using Spherical-deconvolution Informed Filtering of Tractograms (SIFT) to improve the fit between streamline reconstruction and the original dMRI image. The connectome weights were defined by the number of tracks going from one area of the parcellation mask to another, using the Desikan-Killiany atlas[44].

*Functional magnetic resonance imaging acquisition and processing.* The resting state functional MRI (rs-fMRI) scan acquisition was 5 min 8 s using T2$^*$-weighted echo-planar imaging with the following parameters: TR = 2000 ms, TE = 30 ms, 64 × 64 matrix, 20-cm FOV, flip angle = 85, 40 slices, $3.125 \times 3.125 \times 4$ mm$^3$ voxels. Prior to the resting-state functional MRI scan, participants were instructed to close their eyes, not think of anything in particular, and to not fall asleep. Participants were spoken to between each sequence, and prior to each rs-fMRI scan, they were asked if the session could continue. The technicians did not proceed if the participant didn't respond. Functional MRI data were processed using fMRIPrep, an open-source pipeline integrating multiple state-of-the-art fMRI tools into a single software suite[45]. Motion artifact correction and denoising were performed using ICA-AROMA, and susceptibility distortion corrections were performed using the SyN "fieldmap-less" correction method implemented in Advanced Normalization Tools (ANTs). Details on fMRIPrep processing are available in Supplementary Note 1.

BOLD time series of length 308 s (154 time points) were exported in CIFTI format, and the first 18 seconds were discarded to remove initialization transient artifact. In addition, the BOLD time series were pre-processed by removal of any linear trends and constant offsets and passed through a band-pass frequency domain filter with range 0.009 Hz–0.1 Hz. To account for any variations in the fMRI data across the subjects due to physical and physiological aspects of MRI scanning, the BOLD time series per area were z-score normalized for all subjects.

*Serum neurofilament light protein concentration acquisition.* Venous blood samples were collected from participants. Serum NfL concentration was measured using the Human Neurology 4-Plex A assay (N4PA) on an HD-1 Single molecule array

(Simoa) instrument according to instructions from the manufacturer (Quanterix, Billerica, MA).

### Data analysis

*Graph signal processing-based feature extraction.* We modeled brain anatomical areas and connectivity using graph structures whose nodes represent the 66 cortical regions of the Desikan-Killiany atlas and whose edges were their pair-wise connections. Connection weights were computed based on the number of tractography streamlines connecting brain areas, a proxy for alignment and density of fibers in the neuropil, such as axons[46]. For every subject, we used the eigenmodes (i.e. the eigenvalue-eigenvector pairs) of the graph adjacency matrix of the structural connectome to decompose the BOLD signals into low and high graph frequency components according to their conformity to the underlying white matter anatomical network.

Graph Fourier transform (GFT) provides the necessary framework to encode the spatial variability of a graph signal into graph frequencies[47] that are derived from the spectrum of the graph connectivity matrix. In this study, for every subject, we treated the BOLD time series over the brain structural connectome as a graph signal over the brain anatomical network and used the subject-specific spectrum of the brain connectivity matrix to construct graph filters using GFT. The graph filters allowed us to extract different components of the BOLD time series in different brain areas according to their spatial variability. For instance, the energy of the extracted component corresponding to low spatial variability of the BOLD series in a brain area represented the extent to which the BOLD time series in that area conformed to the topology of the brain structural connectome[24].

We next provide the mathematical framework behind the GFT and its application for decomposition of BOLD time series. The brain structural connectome can be represented by an undirected graph $G$ with $n$ nodes, where each node is associated with a distinct brain area. The adjacency matrix of $G$ is given by $\mathbf{A}$, which is an $n \times n$ matrix whose off-diagonal entries is a proxy for the number of axonal connections between different pairs of brain areas. Since $\mathbf{A}$ is symmetric due to inherent limitations of tractography, it can be decomposed as $\mathbf{A} = V \Lambda V^{-1}$, where the eigenvectors of $\mathbf{A}$ form the columns of $V = [v_0, \ldots, v_{n-1}]$ and the eigenvalues of $\mathbf{A}$ are the elements of the diagonal matrix $\Lambda = \mathrm{diag}(\lambda_0, \lambda_1, \ldots, \lambda_{n-1})$, s.t., $\lambda_0 \leq \lambda_1 \leq \ldots \leq \lambda_n$. The formal definition of GFT based on the adjacency matrix is as follows[25]: Given a graph signal $x \in \mathbb{R}^n$ and the adjacency matrix $\mathbf{A} = V \Lambda V^{-1}$, the GFT pair is given by

$$\hat{x} = V^{-1}x, \text{ and } x = V\hat{x}. \quad (1)$$

The eigenvectors of $\mathbf{A}$ form the spectral components of the graph and the eigenvalues of $\mathbf{A}$ form the graph frequencies. The eigenvector-eigenvalue pairs, $(v_k, \lambda_k), \forall k \in \{0, \ldots, n-1\}$, of $\mathbf{A}$ are termed as the eigenmodes of the graph $G$ and are analytically related to the spatial variation of the graph signal (see Supplementary Note 2).

The application of GFT allows us to extract different components of the BOLD time series according to their spatial variation with the help of graph frequency filters of the form

$$F \triangleq \mathrm{diag}(f(\lambda_0), \ldots, f(\lambda_{n-1})), \quad (2)$$

where $f(\lambda_k)$ is the frequency response for the eigenmode $k \in \{0, \ldots, n-1\}$. For a given spatial vector $x$ over the graph, its graph filtered output $y$ is given by

$$y = VFV^T x. $$

As an example, the design of a high pass graph filter based on the adjacency matrix that passes the component corresponding to

10 highest graph frequencies is given by

$$f(\lambda_k) = \begin{cases} 1, & \text{if } k \in \{0, \ldots, 9\} \\ 0, & \text{otherwise} \end{cases}. \quad (3)$$

Although the degree of spatial variability of the BOLD time series with respect to the brain anatomical network varies over a continuum of intermodal 'alignment'[23] or conformity with the brain anatomy, previous studies have demonstrated that the components of graph signals with low or high spatial variability have better and more reliable performance in inference tasks based on neuroimaging data[23,25]. Therefore, in this study, we focused only on the components of BOLD time series with low or high spatial variability.

For each subject, we used the subject-specific graph filter that passed the 10 highest (or lowest) graph frequencies to extract the high (or low) graph frequency components of the BOLD time series. Note that the application of GFT leverages the connectivity of the brain anatomical network to decompose the BOLD time series signal in every TR and therefore, the output obtained after application of a graph frequency filter at any brain area is sensitive to the variation in the BOLD signal with respect to that in the other brain areas[23]. For every area, the application of a low pass graph frequency filter isolates the proportion of its BOLD time series that conforms to the topology of the anatomical network and that of a high pass graph frequency filter isolates the proportion of its BOLD time series that deviates significantly from the topology of the anatomical network. An example of application of high and low pass graph filters on BOLD data is illustrated in Supplementary Fig. 1.

For each brain area, we evaluated the energies of the components with low spatial variability and high spatial variability by calculating the $\ell_2$ norm of the respective graph frequency components of the BOLD time series. Therefore, two features were associated with every brain area for each subject leading to 132 GSP features (66 each from high and low graph frequency analysis) per subject. The group differences between the GSP features in this sample have been reported in our previous study[27].

*Serum NfL level and GSP features.* Prediction and inference form the two paradigms of statistical analysis that provide distinct insights into the relevance of variables depending on the actual modeling goal[48]. Inference helps in isolating individual variables that are significantly associated with the target variable (in this case, serum NfL) whereas prediction driven analysis guides the isolation of variables deemed relevant for predicting the target variable in unseen data. In this study, we aimed to explore the statistical correspondence of GSP features in the context of serum NfL levels in the two cohorts under both statistical paradigms. Due to lack of neuroimaging studies that link specific brain regions with serum NfL, we adopted data-driven approaches to isolate the GSP features that were most relevant to serum NfL from the complete set of 132 features.

_Association between GSP features and serum NfL:_ For the inference paradigm, we adopted a standard linear model based univariate feature selection approach to isolate the GSP features most significantly associated with serum NfL[49]. This approach results in an F-value and a p-value for each GSP feature, whose statistical significance was determined after false discovery rate procedure for correction due to multiple comparisons. Similar approaches have been adopted previously to select the most relevant features from features extracted using GFT of neuroimaging data for various statistical inference tasks[50].

_GSP features as predictors of serum NfL:_ Given the fact that the number of GSP features (132) outnumbered the number of data

samples in both cohorts (20 for HC and 36 for ExPro), we adopted PLSR analysis in our study for prediction paradigm of statistical analysis because of its recommended usage in the neuroimaging literature for scenarios with high multicollinearity among predictors and when the number of predictors outnumber the number of data samples[51,52].

The input and output features were z-score normalized for PLSR analysis. For each group, the PLSR model that fit all the GSP features to serum NfL levels of their respective groups was investigated first. In this context, the number of components for the PLSR model for a given set of features was selected based on the estimated mean square prediction error (MSEP) as the criterion which was evaluated by leave-one-out cross validation. Since the total number of GSP features (132) far exceeded the number of available observations for both groups, the PLSR model with a full set of independent variables was prone to overfitting, which was also confirmed by a nonparametric permutation test of the explained variance $R^2$. The nonparametric permutation test for evaluating the significance of $R^2$ for a PLSR model is described next.

*Nonparametric permutation test for PLSR:* The nonparametric permutation test was carried out by randomizing the serum NfL levels among the subjects and fitting them to the predictors using a PLSR model. The null distributions of $R^2$ were obtained by evaluating the explained variance for 5000 random permutations of the NfL levels. The PLSR models were considered to be overfit for a given set of predictors if the null distribution of the explained variance $R^2$ had a mean >0.5. The statistical significance of $R^2$ values for a PLSR model at a given level was evaluated by counting the number of samples in the corresponding null distribution that exceeded it.

*Variable selection for prediction model:* For both groups, we adopted a variable importance in projection (VIP) based approach for selecting a subset of GSP features that could constitute the PLSR model that fits serum NfL without overfitting[53]. VIP score quantifies the relative importance of each predictor in fitting the PLSR model and was calculated for each predictor based on the PLSR model that fitted the serum NfL levels to 132 predictors for each group. A feature with a high VIP score is typically considered relatively more significant for the prediction performance of the PLSR model[54]. For each group, we varied the threshold of the VIP score and used the features with a VIP score greater than the selected threshold to form the PLSR model with one component. The number of components was set to one due to limited data size.

*Prediction performance based on cross validation:* We evaluated the prediction performance of the models based on leave-one-out cross validation procedure. In both cohorts, for every subject, the non-overfitted model trained on the rest of the subjects with the best prediction performance on that subject's serum NfL was selected as the 'best' model. For both cohorts, the following procedure was followed to calculate the $Q^2$ value for the model. At every instance of cross validation, the serum NfL level for one subject was estimated by the PLS model fitted to the data for the rest of the subjects. The variable selection procedure described above was performed within every instance of cross validation, i.e., VIP scores were evaluated using the serum NfL levels and the GSP features for the subjects in the training set at every cross-validation instance. The set of features for which the PLSR model was not overfitted was chosen to estimate the serum NfL level for the test subject. Therefore, there were 20 PLS models for the HC cohort corresponding to each instance of cross validation. Similarly, there were 36 PLS models for the ExPro cohort corresponding to each instance of cross validation in this cohort. We report the prediction performance from this cross-validation procedure for both cohorts which is quantified by their $Q^2$ values.

For any GSP feature, its frequency of inclusion in the PLS models for prediction of serum NfL for different subjects in cross validation, and similar trends in its respective weights across models, indicates its robustness as a predictor of serum NfL. An overview of the statistical analyses to investigate links between GSP features and serum NfL levels is provided in Fig. 1.

*Clinical and neurological interpretations of GSP features linked with serum NfL.* We used partial correlation, mediation, and moderation analyses to interpret the roles of GSP features that were relevant for serum NfL for both inference and prediction analyses in the two cohorts. Specifically, we investigated whether the GSP features mediated any associations between age, serum NfL, and PAI scores. We also investigated the relationships between the GSP features and structural measures such as cortical thickness and volumes of subcortical regions.

For mediation analysis, we used the mediation toolbox from Wager et al.[55]. The significance of the mediation was established using bootstrapping with 10000 samples. Moderation analysis was conducted based on linear regression and moderation effect was determined based on the significance of the interaction term in the linear model. The reporting of methods and results in this paper adhere to the STROBE guidelines[56].

**Reporting summary**. Further information on research design is available in the Nature Research Reporting Summary linked to this article.

## Results

**GSP features are significantly associated with and predictively relevant for serum NfL in HC and ExPro cohorts**. Our results for univariate feature analysis and PLSR analysis show that a distinct set of GSP features are statistically relevant for serum NfL in HC and ExPro cohorts (Fig. 2). Univariate feature selection yielded 9 GSP features for HC cohort and 24 GSP features for ExPro cohort that had an uncorrected p-value < 0.05 for their respective F-scores (Fig. 2a, d). For HC cohort, low graph frequency features from isthmus cingulate, caudal anterior cingulate, and parahippocampus formed the set of the three most significantly associated GSP features with serum NfL (Fig. 2b), whereas the high graph frequency features were not as strongly associated with serum NfL in this cohort (Fig. 2c). In contrast, we observed that the set of GSP features associated with serum NfL were dominated by high graph frequency features (18 out of 24 with uncorrected p-value < 0.05, see Fig. 2c), specifically in the temporal lobe (transverse temporal, superior temporal and middle temporal areas), lingual and parahippocampus areas in the left hemisphere, and entorhinal in the right hemisphere (Fig. 2f). The low graph frequency features from superior frontal and posterior cingulate areas in the right hemisphere were also significantly associated (Fig. 2e). Therefore, there were significant differences between the set of brain areas associated with serum NfL in the two cohorts. The correlations of all GSP features with serum NfL in the two cohorts are summarized in Supplementary Data 18.

After correction for multiple tests using false discovery rate (FDR) procedure, the low graph frequency feature from isthmus cingulate in the right hemisphere retained statistical significance (FDR corrected p-value = 0.037) for HC cohort (Fig. 2a). For ExPro cohort, FDR correction for multiple tests yielded 7 GSP features significantly associated with serum NfL (6 high graph frequency and 1 low graph frequency, all with corrected p-value = 0.0472) (Fig. 2d). The low graph frequency feature from posterior cingulate area in the right hemisphere was observed to be significant for ExPro cohort (Fig. 2d). These observations clearly indicated that GSP features, which measure

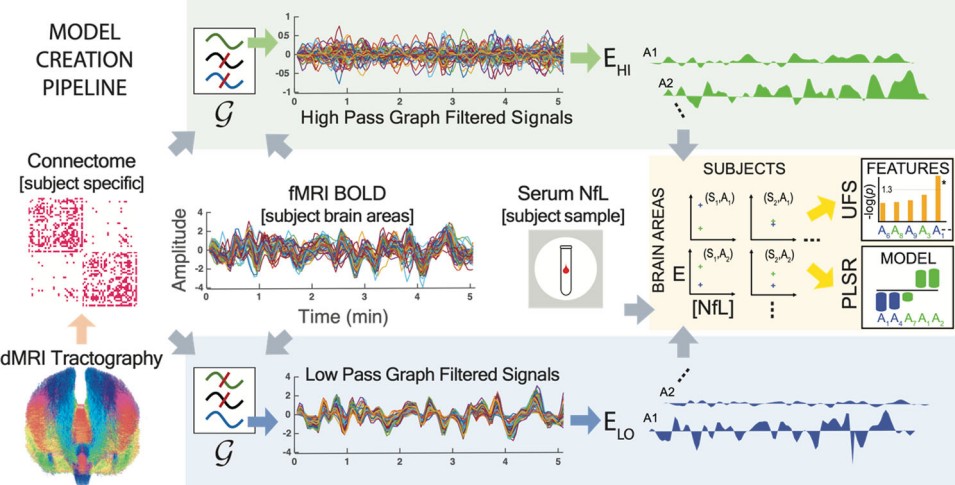

**Fig. 1 Pipeline for statistical analysis for serum NfL levels and GSP-based features.** Brain imaging data (structural (dMRI) and functional (resting state fMRI)), age, and serum NfL levels were recorded for every subject in the cohorts of 20 healthy subjects and 36 former athletes. For each subject, the dMRI and fMRI were pre-processed to extract the structural connectome (in the form of a 66 × 66 adjacency matrix for 66 cortical brain areas) and BOLD signal (in the form of a time series of length 308 seconds at each brain area considered), respectively. Subject-specific graph filters derived from the eigen-decomposition of the structural connectome were constructed and used to extract different graph frequency components of the BOLD time series for all subjects. The energies ($E_{HI}$ and $E_{LO}$, evaluated by $\ell_2$ norm) of the different graph frequency components at different brain areas were investigated (energies of 2 graph frequency components per area for 66 brain areas per subject, i.e., 132 GSP-based predictors) for association with serum NfL levels using univariate feature selection (UFS), and their prediction power evaluated using a rigorous statistical analysis of PLS regression models.

correspondence between functional activity and structure, can link serum NfL levels to specific brain regions in a specific context. Additional experiments also indicated that the association of low graph frequency features from isthmus cingulate and parahippocampal areas in the right hemisphere and high graph frequency features from transverse temporal and lingual areas with serum NfL were different in the two cohorts, thus, implying distinct structure-function coupling profiles for serum NfL in HC and ExPro cohorts (see Supplementary Note 2).

In PLSR analysis, we followed a cross validation procedure for each cohort to investigate the within cohort prediction performance of GSP features. Based on this cross validation, the GSP features predicted 42.26% variance in serum NfL for the HC cohort. The 20 PLS models were evaluated in the leave-one-out cross validation procedure. From these models, we observed that the low graph frequency feature from isthmus cingulate in the right hemisphere was the most robust predictor for serum NfL in the HC cohort, as it was selected as a predictor in the best performing non-overfitted PLS model in every iteration of cross validation. The low graph frequency feature from parahippocampus in the right hemisphere was the next most robust predictor in this cohort. The robustness of different GSP features as predictors for the HC cohort is depicted in the form of a carpet plot in Fig. 3a whose elements indicate the presence of a GSP feature in the best PLS model for every subject. A histogram of the number of subjects for whom the GSP feature was selected as part of the best predicting model is pictorially projected onto a standard cortical surface template in Fig. 3b, c. The distributions of the weights associated with these features are shown in Supplementary Note 3 and Supplementary Fig. 2 (source data available in Supplementary Data 16).

For the ExPro cohort, the GSP features predicted 37.34% variance in the serum NfL based on cross validation. The 36 PLS models were evaluated in the leave-one-out cross validation procedure. From these models, the low graph frequency features from superior temporal and posterior cingulate areas and high graph frequency feature from entorhinal in the right hemisphere, and high graph frequency features from transverse temporal, superior temporal, middle temporal, lingual, and parahippocampus

areas in the left hemisphere were the most robust predictors for serum NfL in this cohort. The robustness of different GSP features as predictors of serum NfL in this cohort is depicted by the carpet plot in Fig. 3d and the histogram is pictorially represented in Fig. 3e, f. The distributions of the weights associated with these features are shown in Supplementary Note 4 and Supplementary Fig. 3 (source data available in Supplementary Data 17). Note that the brain areas revealed to be most robust as predictors of serum NfL in HC and ExPro cohorts (Fig. 3) are consistent with those strongly associated with serum NfL (Fig. 2).

We also investigated the variance explained by a PLS model when trained either on the whole HC cohort or the whole ExPro cohort. Our experiments indicate that each PLS model explains more than 50% of the variance in serum NfL for its respective cohort without overfitting (see Supplementary Figs. 4, 5 and associated discussions in Supplementary Note 5).

We also note that the PLSR models trained on the GSP features from the complete HC cohort did not have any significant predictive ability for serum NfL levels for the ExPro cohort. Conversely, a PLSR model trained on the GSP features from data from all ExPro subjects predicted only 8.9% variance in the serum NfL levels for HC subjects. When the subjects of the two groups were combined to form a single dataset, the GSP features did not explain any significant variance in serum NfL levels. These observations reflect that there was a significant heterogeneity in the brain imaging features extracted from the two cohorts and that, while we could identify GSP features (and their associated brain regions) linked to serum NfL levels within each cohort, the variance in serum NfL levels was not explained by the same set of GSP features across the two populations. This was not unexpected, as TBI is linked to a variety of structural and functional changes in the brain and to elevated serum NfL levels. Moreover, the Kendall's tau coefficient between the VIP scores for the PLSR models (available in Supplementary Data 19) for the two groups was −0.04, indicating that there was no significant consistency between how the GSP features ranked in terms of their relevance to the prediction of serum NfL levels across the two groups. These observations implied that the prediction

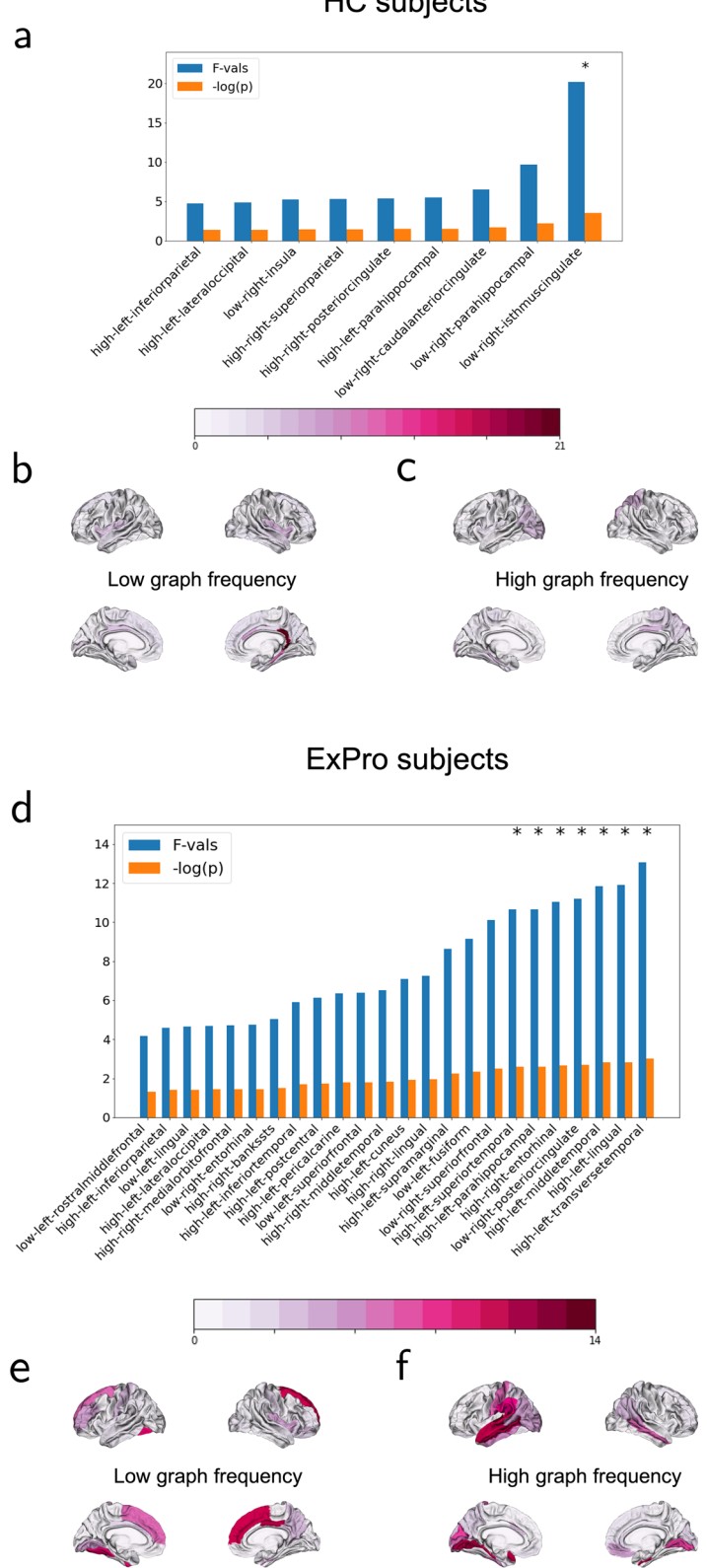

**Fig. 2 Univariate feature selection.** GSP features with uncorrected p-value < 0.05 and corresponding F-values obtained by linear regression based univariate feature selection for **a** HC cohort and **d** ExPro cohort. Higher F-value corresponds to a more significant linear association between GSP feature and serum NfL. Features with corrected p-value < 0.05 after FDR correction for multiple tests are marked with asterisks (*). Panels **b**, **c** plot the F-values for the low and high graph frequency features on a template cortical surface for HC cohort. Panels **e**, **f** plot the F-values for the low and high graph frequency features on a template cortical surface for ExPro cohort.

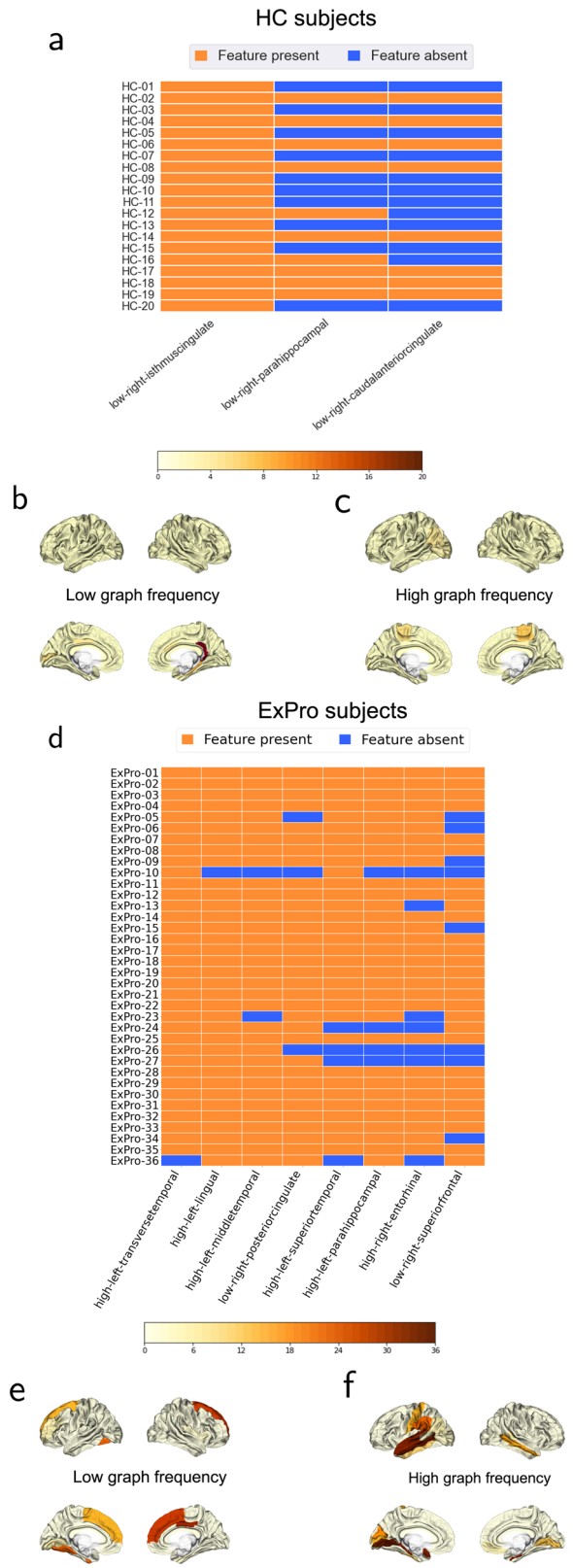

**Fig. 3 Robustness of GSP features as predictors. a**, **d** show the carpet plots illustrating the presence of GSP features in the best performing model in the leave-one-out cross validation process for predicting serum NfL for every subject when it was excluded from the training of model. Each row corresponds to a subject, identified by row ID where HC-X refers to subject X in HC cohort in **a** and ExPro-Y refers to subject Y in ExPro cohort in **d**. The column IDs are associated with GSP features present in the best performing models for 8 or more subjects in the HC cohort in **a** and 18 or more subjects in the ExPro cohort in **d**. Frequency of GSP features associated with different brain areas in the cross-validation models is plotted on the template cortical surface for HC subjects in **b**, **c** and for ExPro subjects in **e**, **f**.

**Table 2 Statistics for mediation analysis between age and serum NfL with GSP feature from right isthmus cingulate as mediator variable for HC subjects.**

|  | Coefficient | Std. error | p value (uncorrected) |
|---|---|---|---|
| Path a | −0.05 | 0.01 | 0.0038 |
| Path b | −2.20 | 0.84 | 0.0001 |
| Path c' (adjusted effect) | 0.16 | 0.06 | 0.0292 |
| Mediation (ab) | 0.11 | 0.03 | 0.0014 |
| Path c (total effect) (age -> serum NfL) | 0.2658 | 0.0646 | 0.0012 |

prediction paradigms of analysis. In this context, the low graph frequency features from isthmus cingulate, superior frontal, and posterior cingulate areas in the right hemisphere, and high graph frequency features from transverse temporal, superior temporal, middle temporal, lingual, and parahippocampus areas in the left hemisphere and entorhinal in the right hemisphere were chosen. Recall that high and low graph frequency features for a brain region relate to the coupling between the functional activity and underlying white matter anatomy (see Supplementary Fig. 1 for an example of different graph frequency components of BOLD data). Since some of these features also showed group differences in terms of their associations with serum NfL in the two cohorts (see Supplementary Note 2), we conjectured that these GSP features may have distinct characteristics in the two cohorts, and therefore we analyzed them in both cohorts for subsequent experiments. However, the association of these GSP features with serum NfL in the previous experiments did not imply their relevance to association between serum NfL and age, personality scores or structural measures investigated in the analyses that follow.

*GSP features complement and are independent of age in predicting serum NfL levels.* In this set of experiments, we explored the relationships among age, serum NfL levels and GSP features. We hypothesized that aging was a causal factor that affected a subset of GSP features and serum NfL levels across the two groups, and therefore it was treated as a confounding variable. We observed that serum NfL was significantly correlated with the low graph frequency GSP features from isthmus cingulate ($\rho = 0.5425$, FDR corrected $p$-value = 0.04) in the right hemisphere after correction for age, which indicated that this brain area had a significant association with the serum NfL levels independent of age in the HC group.

The statistics for mediation analysis for testing the mediation of the low graph frequency feature from isthmus cingulate in the right hemisphere on association between age and serum NfL are summarized in Table 2, where the coefficient of path "a" is a

performance of PLS models was not transferable across the groups (see Supplementary Note 6 for additional discussions).

**Clinical and neurological interpretations of GSP features**. We focused our subsequent analysis on the set of features that were deemed of interest by our experiments under both inference and

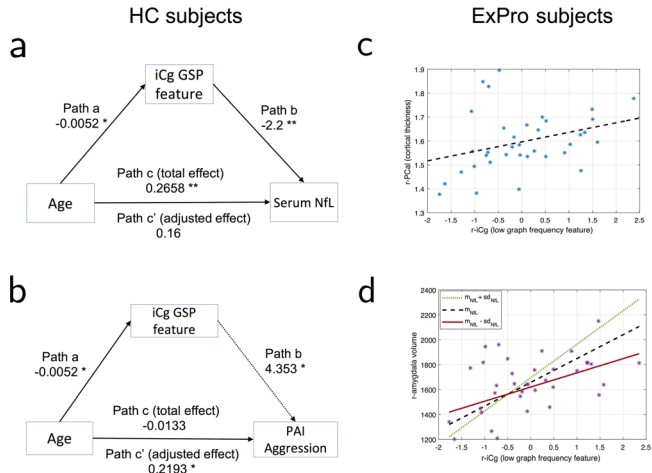

**Fig. 4 Clinical and neurological interpretations for GSP features based on mediation and moderation analyses.** (** *p* value < 0.01, * *p* value < 0.05, all *p* values were FDR corrected for multiple comparisons) **a** Mediation path diagram with GSP feature as mediator, age as predictor and serum NfL level as the dependent variable. Partial mediation effect (significance determined by *p* value of path "ab") was observed for low graph frequency feature for isthmus cingulate in right hemisphere (low r-iCg) for HC cohort. **b** Mediation path diagram for HC subjects with age as predictor, PAI Aggression score as dependent variable and iCg GSP feature as the mediator. The indirect path of age on aggression score through the mediator was opposite to the direct effect (see Table 3 for details). Therefore, this GSP feature was observed to have a "suppression" or inconsistent mediation impact on the relationship between age and aggression score for HC subjects. **c** This panel plots cortical thickness of pericalcarine area in the right hemisphere versus the low graph frequency feature from isthmus cingulate (iCg) area in right hemisphere for ExPro cohort. This GSP feature was positively correlated with the cortical thickness of pericalcarine area in both hemispheres for ExPro subjects ($\rho$ = 0.482, FDR corrected p-value = 0.0238) when adjusted for age. **d** This panel illustrates moderation analysis for serum NfL on the relationship between the volume of right amygdala and r-iCg feature in ExPro subjects. $m_{NfL}$ and $sd_{NfL}$ are the mean and standard deviation of serum NfL. Larger slope for $m_{NfL} + sd_{NfL}$ line indicates that subjects with high serum NfL had a higher volume in right amygdala as a function of the r-iCg feature as compared to those with average serum NfL ($m_{NfL}$ line) and low serum NfL ($m_{NfL} - sd_{NfL}$ line).

measure of the association between age and the mediator variable, i.e., GSP feature from isthmus cingulate, the coefficient of path "b" is a measure of association between serum NfL and the mediator variable, the coefficient path "c" is a measure of the direct effect of age on serum NfL when adjusted for the mediator variable, and the coefficient of path "ab" is a measure of the indirect effect of age on serum NfL via the mediator variable.

Mediation analysis revealed that the low graph frequency from the right isthmus cingulate had a significant partial mediating effect (based on significance of path 'ab', FDR corrected *p* value = 0.0098) on the association between age and serum NfL (see Fig. 4a). Clearly, all paths and the mediation effect were statistically significant (*p* value < 0.05, Table 2) and the coefficient for adjusted effect, i.e., path "c'" was smaller than that for total, unmediated effect, i.e., path "c'" between age and serum NfL. From the observations in the mediation analysis and partial correlation analyses, we conclude that the low graph frequency feature from isthmus cingulate in the right hemisphere captured age-independent variation in serum NfL for HC group and also partially explained the association between serum NfL and age in HC subjects.

**Table 3 Statistics for mediation analysis between age and PAI Aggression score with GSP feature from right isthmus cingulate as mediator variable for HC subjects.**

| | Coefficient | Std. error | *p* value (uncorrected) |
|---|---|---|---|
| Path a | −0.052 | 0.0139 | 0.0035 |
| Path b | 4.353 | 1.217 | 0.0032 |
| Path c' (adjusted effect) | 0.2193 | 0.0806 | 0.0029 |
| Mediation (ab) | −0.2326 | 0.0961 | 0.0122 |
| Path c (total effect) (age->PAI Aggression score) | −0.0133 | 0.0988 | 0.9761 |

For the ExPro group, we observed that the low graph frequency feature associated with the posterior cingulate area in the right hemisphere did not have a significant correlation (*p* value > 0.05) with serum NfL levels when age was used a confounding variable. Furthermore, mediation analysis revealed no mediating effect for these GSP features on the relationship between age and serum NfL in the ExPro group. However, serum NfL levels had significant partial correlation with the high graph frequency features associated with the parahippocampus ($\rho$ = 0.4564, FDR corrected *p* value = 0.009), left transverse temporal area ($\rho$ = 0.475, FDR corrected *p* value = 0.016) and right entorhinal area ($\rho$ = 0.407, FDR corrected *p* value = 0.028) when corrected for age. Furthermore, the GSP features from entorhinal in the right hemisphere and transverse temporal from the left hemisphere showed evidence of partial mediating effects on the relationship between age and serum NfL (see Supplementary Tables 1, 2). Therefore, our results suggest that the GSP features captured the variation in serum NfL levels in both groups that were independent of aging.

Thus far, we have established the utility of GSP features in predicting serum NfL in the two groups. However, our experiments indicate that the predictive performance of the group-specific models in Fig. 3 does not translate across groups. Since aging was a common factor that had an impact on serum NfL for both groups, we also performed an exploratory analysis aimed at building a model that had a significant prediction performance for serum NfL levels across the two groups by leveraging GSP features associated with age (see Supplementary Note 7 and associated results in Supplementary Tables 3, 4). In these experiments, we observed that a linear model with the low graph frequency feature from isthmus cingulate in the right hemisphere as the predictor and serum NfL as the response variable predicted a significant amount of variance in serum NfL for HC group (35.9%) when the model was trained using data from ExPro group.

*GSP features in NfL prediction models are associated with personality scores in healthy controls.* In this set of experiments, we evaluated the association of high and low graph frequency features observed to be relevant for serum NfL in the two cohorts with the PAI, which assesses a subject's personality and psychopathology[56]. We focused our subsequent analysis only on the personality scores that were significantly different in the two groups and considered valid for TBI, namely the Aggression and Depression subscales. We used raw scores for our analysis, and similar observations were made when scores were T-normalized. We did not observe significant associations between GSP features and Depression subscale scores. In contrast, our experiments showed several differences in the association of PAI scores on Aggression subscale (referred to as PAI Aggression score in the rest of the paper) with GSP features in the two cohorts. Therefore, in this section we present the results only from aggression.

We observed that PAI Aggression score was positively correlated with the low graph frequency feature from isthmus cingulate ($\rho = 0.7052$, $p$ value $= 7.45e{-}4$) for HC subjects but not for ExPro subjects ($\rho = 0.1873$, $p$ value $> 0.05$) when age was used as a covariate. These observations indicated that higher PAI Aggression score was associated with a stronger structure-function coupling in the isthmus cingulate area in the right hemisphere for HC cohort. Notably, for ExPro subjects, we also observed a statistically significant partial correlation between serum NfL and PAI Aggression scores after correction for age ($\rho = -0.44$, $p$ value $= 0.0076$). No relationship was observed between serum NfL and PAI aggression scores in HC subjects. The lack of correlation between serum NfL and PAI aggression score in HC cohort as compared to the significant positive correlation among the ExPro cohort merits further neurological exploration, which was not the focus of this work. However, the concurrently observed decreased correlation between the GSP feature from isthmus cingulate and PAI aggression score in ExPro cohort in our experiments corroborates the relevance of GSP features in understanding the brain mechanisms behind aggression and differentiation of serum NfL levels in the two populations.

We observed that the PAI Aggression score had no correlation with age (absolute correlation $<0.05$) in HC subjects. However, when the correlation was corrected for the low graph frequency feature from isthmus cingulate, we observed a positive partial correlation between age and PAI Aggression score in this cohort ($\rho = 0.4866$, uncorrected $p$ value $= 0.0346$) which indicated potential suppression effect of the GSP feature from isthmus cingulate, i.e., reflected the canceling out of the positive direct effect of age on aggression score through an indirect path linking aging to the GSP feature from isthmus cingulate. To test this interpretation, we performed mediation analysis for HC subjects with age as a predictor, PAI Aggression score as the dependent variable and low graph frequency feature from isthmus cingulate in right hemisphere as mediator. We observed that the low graph frequency feature from isthmus cingulate indeed had a significant inconsistent mediating effect or suppression effect (uncorrected $p$ value for indirect path $= 0.0122$, FDR corrected $p$ value $= 0.0787$) on the association between age and PAI Aggression score (see Fig. 4b and Table 3).

While exploring the causal association of GSP features with the personality traits in the two cohorts was not the focus of this paper, our observation regarding the suppression effect on aggression score by a GSP feature which was negatively correlated with serum NfL indicates that there may exist a novel pathway using GSP features to assess interplay between neurodegeneration and personality and could contribute to the understanding of psychopathology.

### GSP features are associated with structural measures.
We explored the association of the GSP features with structural measures including volumes and thickness of different cortical regions.

*Cortical thickness:* There was no statistically significant difference in the cortical thickness of different regions in the two cohorts. We observed that for the HC cohort, the thickness of pericalcarine region in the right hemisphere was negatively correlated with serum NfL (partial correlation with correction for age, $\rho = -0.622$, $p$ value $= 0.003$), which was in line with our hypothesis that cortical thickness may be negatively correlated with serum NfL. The significant correlation after correction for age indicated that this association may not be driven by aging in HC cohort. We did not observe any other cortical thickness measures to be correlated with serum NfL for ExPro or HC subjects at 0.01 significance level (uncorrected). Interestingly, cortical thinning of pericalcarine region is linked with impulsive

and risky tendencies in the existing studies[39] which characterize the behavioral impacts of TBI. Therefore, we further investigated the association of the pericalcarine thickness with serum NfL and GSP features in both cohorts.

For HC subjects, a non-significant positive correlation was observed between the cortical thickness of pericalcarine region in the right hemisphere and the low graph frequency feature from isthmus cingulate in the right hemisphere ($\rho = 0.357$, $p$ value $= 0.1335$). Since serum NfL is a marker of neurodegeneration, we hypothesized the cortical thickness of pericalcarine to be a causal factor for variation in serum NfL levels. The mediation analysis hinted at a partial mediation effect for the low graph frequency region from isthmus cingulate in the right hemisphere on the association between the cortical thickness of pericalcarine region and serum NfL (see Supplementary Note 8 and Supplementary Table 5).

The low graph frequency feature from isthmus cingulate in the right hemisphere was more significantly correlated with the cortical thickness of pericalcarine area in right hemisphere for ExPro subjects ($\rho = 0.482$, FDR corrected $p$-value $= 0.0238$) than the HC subjects when adjusted for age (see Fig. 4c). Therefore, the association of pericalcarine thickness with the low graph frequency feature from isthmus cingulate in both cohorts must be further explored. We also remark that we observed significant correlation between the low graph frequency feature from caudal anterior cingulate in the right hemisphere and cortical thickness of postcentral area in both hemispheres for ExPro subjects (Supplementary Note 10 and Supplementary Fig. 6). This GSP feature was the third most significantly associated GSP feature with serum NfL for HC cohort (Fig. 2a).

*Subcortical volumes:* Volume of amygdala in right hemisphere had significant negative correlation with age ($\rho = -0.6574$, $p$ value $= 1.37e\text{-}5$) and serum NfL ($\rho = -0.4194$, $p$ value $= 0.0109$) for ExPro subjects but not for HC subjects. We further observed that the volumes of right amygdala did not have a significant correlation with serum NfL for ExPro subjects when age was used as a covariate. Volume of amygdala in the left hemisphere was significantly associated with age ($\rho = -0.3926$, $p$ value $= 0.0179$) for ExPro subjects but not for HC subjects. No significant association with serum NfL was observed for volume of amygdala in the left hemisphere for either cohort. We hypothesized that aging was the driving factor behind the change in volume of amygdala in both hemispheres for ExPro subjects. We focused our subsequent analysis only the volume of amygdala in the right hemisphere since it was observed to be relevant for serum NfL in ExPro subjects.

We tested the association of aging related GSP features in the cingulate gyrus with the volumes of right amygdala in both cohorts. Our experiments revealed that the low graph frequency feature from isthmus cingulate in the right hemisphere had a significant positive correlation with the volume of right amygdala for ExPro subjects ($\rho = 0.4597$, uncorrected $p$ value $= 0.0048$, FDR corrected $p$ value $= 0.0336$).

Our analysis revealed that serum NfL moderated the interaction between the volume of right amygdala and the low graph frequency feature from isthmus cingulate in ExPro subjects (see Fig. 4c and Table 4). Specifically, the rate at which volume of right amygdala increased as a function of the low graph frequency energy in isthmus cingulate for subjects increased with increasing serum NfL levels in the population. Notably, age did not have a similar moderating effect as serum NfL, thus disassociating the interpretation of serum NfL levels from aging in ExPro cohort. Similar observations were not present in the context of HC subjects. Interestingly, reduced amygdala volume has been linked with increasing aggressive tendencies in both healthy subjects and pathological contexts[31,40].

**Table 4 Statistics for moderation analysis for interaction between GSP feature from right isthmus cingulate and right amygdala volume for ExPro subjects.**

|  | Coefficient | Std. error | p-value (uncorrected) |
|---|---|---|---|
| Age | −11.092 | 25.51 | 0.0003 |
| Serum NfL | 8.0921 | 7.35 | 0.27 |
| r-iCg GSP feature | 30.249 | 27.869 | 0.28 |
| Interaction (serum NfL X r-iCg feature) | 16.495 | 5.51 | 0.0054 |
| Intercept | 1679.6 | 65.84 | 7.07e−35 |

Additional experiments also showed significant association of the volume of choroid plexus with serum NfL and the low graph frequency feature from posterior cingulate in the right hemisphere in the ExPro cohort. We have discussed these observations in Supplementary Note 9 and Supplementary Table 6. While the analysis of choroid plexus is relevant in the context of studies of neurodegenerative diseases[57,58], this structure is not known to be relevant to the functional connectome, and so the spuriousness of these relationships could not be ruled out.

## Discussion

The current study and results add a new dimension to the analysis of serum NfL levels in the context of traumatic brain injury and, by association, neurodegenerative diseases by demonstrating that brain activity patterns decomposed over the brain's structure are partially explanatory and predictive of serum NfL levels in two distinct cohorts. For each cohort, we observed a convergence between the findings from inference and predictive paradigms of the statistical analyses. Our results showed that the low graph frequency feature from isthmus cingulate in the right hemisphere (which is positively correlated with structure-function coupling in this area) had the strongest association and most robust predictive performance for serum NfL in healthy controls. In contrast, the low graph frequency features from superior frontal and caudal anterior cingulate in the right hemisphere and high graph frequency features from temporal lobe and lingual in the left hemisphere (which are negatively correlated with structure-function coupling in the corresponding areas) had both the strongest association and predictive relevance for serum NfL in the ExPro cohort. Therefore, our analyses clearly established the significance and heterogeneity of neuroimaging biomarkers associated with serum NfL in two different populations with similar serum NfL levels.

Our findings further allow the possibility of mapping variation in brain structure and functional networks beyond what is expected by normal aging onto the variation in serum NfL levels among different cohorts of patients. For both healthy controls and former athletes, we investigated whether the GSP features mediated the relationships between age and serum NfL. Age-related atrophy in both cortical and white matter is well established[59,60]. Therefore, it was expected that age had predictive capacity of serum NfL levels. Moreover, existing studies show that different brain areas may exhibit different levels of coupling between the brain's functional activity and structure[28]. Analyzing how white matter alterations during normal aging characterize the variation in the conformity of brain area BOLD signals to underlying anatomical connectivity was beyond the scope of this study. However, such an analysis is warranted to examine the specific roles different brain areas have in the association between their GSP features and serum NfL levels in the PLSR model, and to disambiguate the causal ordering of changes in conformity of

BOLD to brain connectivity. Accelerated and distinct changes in brain networks are often observed in the context of different neurological disorders. For instance, the limbic system has been shown to be affected in TBI[61] and various other disease contexts, such as Parkinson's disease[62], dementia[63], and depression[64]. Our results show an association of multiple brain areas of the limbic system, such as the cingulate gyrus, the parahippocampal gyrus and the amygdala with serum NfL levels, which in turn varies in the aforementioned pathological contexts. Interestingly, our results show that regions in the cingulate gyrus exhibit different characteristic relationships with serum NfL levels and aging in the two groups. This study is the first to our knowledge that combines structural connectivity, functional networks and serum NfL levels to extend today's blood biomarkers towards including neuroimaging features. It follows then that, by way of example, if our results are shown to generalize for other cohorts, the diagnosis of neurodegeneration in a new patient might be facilitated by identifying those GSP features obtained from their neuroimaging data that are most strongly associated with their serum NfL levels and then determining if these features constitute a normal age-associated correlation or a pathological association.

Although not the primary focus of our paper, our analyses of scores from the PAI and structural measures further supplemented the roles of GSP features from the limbic system beyond serum NfL prediction to clinical observations in the two cohorts. In healthy controls, the low graph frequency energy from isthmus cingulate was associated with the PAI aggression subscale, which reflects the utility of this GSP feature in further substantiating the potential for temper and aggressive behavior-related complications in clinical treatment planning[65]. Our statistical analyses show that the GSP feature from isthmus cingulate reflected a suppressing effect on the causal relationship between age and aggression in HC subjects. Specifically, our mediation analysis in HC subjects revealed that the low graph frequency feature from isthmus cingulate (1) decreases with age and (2) is directly proportional to an increase in aggressive behavior when regressed to PAI Aggression score jointly with age, therefore implying a negative effect on aggression score by age via the path through the low graph frequency feature from isthmus cingulate in the right hemisphere (i.e., paths "a" and "b" in Fig. 4b). This negative effect through the mediated path negated the direct, increasing effect of age on aggression in the HC cohort (path "c'" in Fig. 4b) and resulted in a non-significant total effect of age on aggression score. The above phenomenon was not statistically significant in ExPro subjects.

Interestingly, the GSP feature from isthmus cingulate was positively correlated with the volume of right amygdala in ExPro subjects, and statistical analysis showed that this association was moderated by serum NfL levels, i.e., higher serum NfL level implied a steeper positive correlation. The amygdala is a critical brain region responsible for processing emotional responses to sensory stimuli in humans[66] and is known to suffer volume loss post TBI[67]. Studies also reported that the volume of right amygdala plays a modulating role on the aggressive trait in healthy subjects[31] and reduction in amygdala volume is associated with increasing aggressive behavior, which is of relevance to various clinical contexts[40]. Since right amygdala volume was negatively correlated with age and serum NfL in ExPro cohort but not HC cohort, we propose that our findings suggest the mechanism of modulating aggression may be overwhelmed in ExPro cohort. This conjecture is supported in part by a larger slope of variation in the GSP feature-right amygdala volume curve for high serum NfL in Fig. 4d. Furthermore, if the previous finding that a smaller right amygdala volume is a cause of increase aggression also applies to the ExPro cohort in our study, our observations suggest that the effect of aging on increasing

aggression is compounded by more significant physiological changes in right amygdala in ExPro subjects. We hypothesize that the distinction in the characteristics of low graph frequency energy in isthmus cingulate in the two cohorts coupled with physiological changes in right amygdala could partly predict the higher aggressive tendencies in the ExPro cohort as compared to the HC cohort. Note that we also observed a statistically significant negative correlation between serum NfL and PAI Aggression scores, which was independent of aging and counterintuitive to the hypothesized effect of reduction in right amygdala volume on aggressive behavior in the ExPro cohort. This observation indicates the presence of a distinct suppression mechanism on the aggressive behavior in the ExPro cohort involving serum NfL that must be explored further.

The cortical thickness of the pericalcarine region in the right hemisphere is another structural measure linked to aggressive tendencies. Specifically, reduced pericalcarine thickness has been reported to be linked with impulsive personality traits and engagement in risky behaviors[39], which are known possible after-effects of TBI, potentially persisting for years[67,68]. Therefore, the observed negative correlation between serum NfL levels and pericalcarine thickness and positive correlation between the low graph frequency energy from isthmus cingulate in the right hemisphere for HC subjects were along expected lines. Serum NfL is a marker for neurodegeneration and, therefore, the negative correlation between serum NfL and pericalcarine thickness in HC subjects was also expected. For both HC and ExPro cohorts, we observed a positive correlation between pericalcarine thickness and the low graph frequency energy from isthmus cingulate in the right hemisphere. However, serum NfL did not have a significant correlation with pericalcarine thickness in the ExPro cohort. These observations suggest that the loss in pericalcarine thickness may have contributed significantly to serum NfL levels only in HC subjects, indicating that other factors may be at play in ExPro cohort aside from normal aging.

Personality traits are known to correlate with neuropsychiatric symptoms[69] and their role in predicting cognitive health is an ongoing area of research[70]. Therefore, our observations that GSP features from isthmus cingulate predict serum NfL levels and associate with structural measures linked to aggressive tendencies provide proof-of-concept that using our methodology, GSP features can refine blood biomarkers of neurodegeneration and augment their interpretation in terms of personality traits and cognition by enhancing understanding of causal pathways from structural measures alone. The emphasis on the right isthmus cingulate among our results needs to be replicated in a larger cohort of subjects with a variety of neurological conditions affecting the brain.

Additional analyses show that the GSP features in the limbic system were correlated with several structural measures in the ExPro subjects that may be indicative of pathological outcomes reported in the existing studies. For instance, our experiments showed that for ExPro subjects, the low graph frequency feature from right caudal anterior cingulate region was associated with cortical thickness of postcentral regions in both hemispheres, which has been revealed as a metric of interest in veterans with a history of TBI and post-traumatic stress disorder[71]. Furthermore, additional experiments showed that the low graph frequency feature of posterior cingulate is correlated with the volume of left choroid plexus in ExPro subjects. The choroid plexus produces CSF and is part of the post injury mechanism to promote healing and stabilize cognitive processes[71,72]. In a wider context, the degeneration of structure and function of choroid plexus can contribute to cognitive deterioration in neurodegenerative diseases[57,58]. Although the existing studies suggest the importance of exploring choroid plexus as a region of interest in different contexts, we remark that the results involving the choroid plexus volume in our study could potentially be affected by inaccuracy of choroid plexus segmentation by the Freesurfer package, particularly in the ExPro cohort. We conjecture that if valid, the changes in left choroid plexus volume observed in our experiments are linked to aging related variation in serum NfL in ExPro subjects and these findings merit further investigation.

In summary, our results suggest that both low and high graph frequency features jointly provide key insights into the variance in serum NfL levels among healthy control subjects and former athletes with a history of concussion than does age alone. Interestingly, deviations in the signs of the associations between serum NfL levels and the high graph frequency energy of the right paracentral area and low graph frequency energy of the left pericalcarine area from their hypothesized behavior reaffirms that structural anatomy may be an incomplete determinant of functional activity in brain networks[73,74], which our results indicate even for resting state brain activity (see Supplementary Note 2). Relationships of GSP features with volume measures and cortical thickness of various brain regions revealed several associations of GSP features with metrics of significant interest among clinical and pathological studies of TBI. Further analysis may provide additional information regarding changes that brain networks normally undergo with age versus those due to other factors such as concussion, thus leading to variations in serum NfL levels.

## Data availability

The neuroimaging data that supports the findings of this study is subject to confidentiality agreement and the patients have not consented to public release of their data. Access to the neuroimaging dataset can be requested to M.C.T. (Carmela.Tartaglia@uhn.ca). GSP features extracted from neuroimaging data and serum NfL levels for HC and ExPro cohorts that support the results in Figs. 2, 3 are provided as Supplementary Data 1,2,3, and 4. The age data for HC and ExPro cohorts are available in Supplementary Data 5 and 6. PAI aggression scores for HC and ExPro cohorts are available in Supplementary Data 7 and 8. Cortical thickness data for HC and ExPro cohorts are available in Supplementary Data 9 and 10. Source data files for Fig. 2a–c are available in Supplementary Data 12, for Fig. 2d–f are available in Supplementary Data 13, for Fig. 3a–c are available in Supplementary Data 14, and for Fig. 3d–f are available in Supplementary Data 15. Source data for Fig. 4a is available in Supplementary Data 1, 3, and 5. Source data for Fig. 4b is available in Supplementary Data 1,3, and 7. Source data for Fig. 4c is available in Supplementary Data 2 and 10. Source data for Fig. 4d is available in Supplementary Data 2, 4, and 11.

## Code availability

The code to reproduce the analysis and figures in this paper is written in Matlab, python and R. The univariate feature selection was performed using scikit-learn package in python[49]. PLSR analysis and moderation analysis were performed using inbuilt functions in Matlab. Mediation analysis was performed using mediation toolbox from Wager et al.[55]. The brain surface plots in Figs. 2 and 3 were generated using 'fsbrain' package in R[75]. The code for univariate feature selection and PLSR analysis is available online[76] at https://doi.org/10.5281/zenodo.5651347.

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

## Acknowledgements

We would like to thank Mozhgan Khodadadi for her help on the administration of this clinical research project. KB is supported by the Swedish Research Council (#2017-00915), the Alzheimer Drug Discovery Foundation (ADDF), USA (#RDAPB-201809-2016615), the Swedish Alzheimer Foundation (#AF-742881), Hjärnfonden, Sweden (#FO2017-0243), the Swedish state under the agreement between the Swedish government and the County Councils, the ALF-agreement (#ALFGBG-715986), and European Union Joint Program for Neurodegenerative Disorders (JPND2019-466-236). HZ is a Wallenberg Scholar supported by grants from the Swedish Research Council (#2018-02532), the European Research Council (#681712), Swedish State Support for Clinical Research (#ALFGBG-720931), the Alzheimer Drug Discovery Foundation (ADDF), USA (#201809-2016862), and the UK Dementia Research Institute at UCL.

## Author contributions

Conception and design of the work: S.S., J.R.K., M.C.T. Data collection: F.T., M.G., R.W., D.J.M., K.B., H.Z., L.G.D., R.G,. B.C. Data analysis: S.S,. L.G.D. Interpretation of results: S.S., S.N,. J.R.K,. M.CT. Drafting the article: S.S., S.N., J.R.K. Critical revision of the article: J.R.K., C.T., M.C.T. Funding acquisition: J.R.K., M.C.T., C.T. Software contribution: S.S., S.N. Administration: R.G., B.C.

## Competing interests

The authors declare the following competing interests: MCT has served as at advisory board for Biogen, Denali, and Roche. KB has served as a consultant, at advisory boards, or at data monitoring committees for Abcam, Axon, Biogen, JOMDD/Shimadzu, Julius Clinical, Lilly, MagQu, Novartis, Roche Diagnostics, and Siemens Healthineers, and is a co-founder of Brain Biomarker Solutions in Gothenburg AB (BBS), which is a part of the GU Ventures Incubator Program (all outside the present paper). HZ has served at scientific advisory boards for Denali, Roche Diagnostics, Wave, Samumed, Siemens Healthineers, Pinteon Therapeutics and CogRx, has given lectures in symposia sponsored by Fujirebio, Alzecure and Biogen, and is a co-founder of Brain Biomarker Solutions in Gothenburg AB (BBS), which is a part of the GU Ventures Incubator Program (outside submitted work). All other authors do not have any competing interests to declare.
