## [Peer Review File · Communications Medicine]

Reviewers' comments:

Reviewer #1 (Remarks to the Author):

I first wish to give my best encouragements to the authors regarding this work, which fruitfully exploits Graph Signal Processing to study individual differences in function-structure relationships, by combining anatomical connectivity with BOLD signals using the GSP framework. The extracted GSP features based on signal energy in either high or low frequency are then related to serum Nfl levels using a very wide range of statistical inferences, as well as a proposal regarding multivariate analysis using PLSR. The GSP part is very well done and follows the best standards in this (emerging) field. Again, I was pleased to see GSP methods adapted to a new neuroscientific question. However, I have many comments regarding the rest of the analysis, which I think in the current state suffers from important limitations (see my detailed comments below). In addition, some of the parts in the discussion seem a little too speculative, as the results presented in this paper only concerns a limited sample size. Some of my following questions may seem very basic but they are critical questions, that are important both for estimating the validity of the PLSR analysis, but also for reproducibility of such results. I will not comment much on the inferential statistics as well as on the interpretation part of the paper, as I think the methodology already needs to be revised.

1 - A general feedback on the analytical approach of the paper. In its current form, the PLSR may appear insufficient for an expert in statistical learning, while being confusing for a reader that is not very knowledgeable on multivariate methods. First, for anyone familiar with machine learning, it is very surprising to explicitly present a set of analysis that extremely overfit on the training set, to then explain how to select features that don't overfit on the test set. The overall sample size is also extremely limited to perform PLSR. Some of the next questions in my review address alternative solutions to avoid this caveat. In the current manuscript, after the proposed PLSR analysis, the following and largest part of the analysis is done using more classical inference based methods, using correlations and mediation analysis (which are more suited for this sample size). As a result, when trying to see the bigger picture in the approach taken, I have the impression that PLSR is solely performed in order to select a small set of relevant GSP features. PLSR was done using cross validation in order to enhance the predictive power, but in the rest of the analysis, inference based statistics are used with bootstrapping. The mix of a prediction based and inference based methods can be difficult to interpret, and might be considered a bit circular in the opinion of some readers and some experts in this community (see for example the series of papers by Danilo Bzdok on prediction versus inference). As a consequence, I am wondering what the authors think about the articulation of the different analytical approach used in the paper. For instance, how much of the analysis was hypothesis based versus data driven ?

2 - How many components were used for PLSR ? I have checked multiple times and this was never mentioned in the paper, however this is a major defining feature in PLSR. The number of components will directly influence the capacity of the model to find a common space to link the inputs with the outputs. In addition, have the authors tested to vary the number of components ?

3 - Are the input and output features scaled or normalized ?

4 - Such questions would easily be solved if the authors could release the code for review ; I have seen that the authors plan to release the code upon acceptance, but when it comes to such elaborated analysis, having the code for review would enable a much more efficient review process

(at least for me!), as it would answer 90% of the questions I ask here.

5 - From Figure 2 it is really not obvious which ones are the "best" PLSR models. On line 240-241, the best model for ExPro is has 52% R2 but when looking at figure 2B, there are a few models around 0.5, and also models with higher R2 values, more than 0.6. Why aren't those last models better ? In Figure 2A, however, the "best" model corresponds to the description in the text (58.6% R2) as there are only two significant models. Therefore please clarify the choice of the "best" model for ExPro. This clarification is particularly important as it will decide which features are eventually kept for healthy subjects.

6 - The overall methodology for selecting small sets of features (4 or 5 out of 132 GSP features!) using PLSR does not seem very sound to me considering the small number of examples. Because of the small sample size, the fact that filtering features with VIP scores larger than 1 still yields overfitting may indicate that the problem does not come from collinearity in features (as suggested by the authors), but from a low explained variance in many features. In addition, when setting very high thresholds in VIP, there are only 4 or 5 features left, which indicates that those are the 4 or 5 features that have a linear relationship with NFL scores in the first place. Could the authors justify better the PLSR approach performed here ?

7 - Alternatively, because of all the drawbacks described in the previous comments, I suggest using a much simpler feature selection method, based on univariate statistics, such as f-tests (anova directly with the output variables), or recursive feature elimination (which can also be done with cross validation), all that is explained here : https://scikit-learn.org/stable/modules/feature_selection.html#univariate-feature-selection . Notably, this feature selection method has been successfully applied to select relevant GSP features in previous work from my group, for classification (Menoret et al. 2017, Brahim & Farrugia 2020) and regression (Pilavci & Farrugia 2019, ICASSP). Such a method might be more convincing and easy to understand for the reader, as currently the PLSR setup seems a little convoluted, and critical details are missing (such as the number of components, the implementation used, ..see my earlier comments). Additionally, using univariate statistics for feature selection will be easier to articulate with the rest of the analysis performed, which are mostly based on inferential statistics. Feel free to include any of the following citations if you find this approach relevant :

Ménoret, M., Farrugia, N., Padeloup, B., & Gripon, V. (2017). Evaluating graph signal processing for neuroimaging through classification and dimensionality reduction. In 2017 IEEE Global Conference on Signal and Information Processing (GlobalSIP) (pp. 618-622). IEEE.

Pilavci, Y. Y., & Farrugia, N. (2019). Spectral graph wavelet transform as feature extractor for machine learning in neuroimaging. In ICASSP 2019-2019 IEEE International Conference on Acoustics, Speech and Signal Processing (ICASSP) (pp. 1140-1144). IEEE.

Brahim, A., & Farrugia, N. (2020). Graph Fourier Transform of fMRI temporal signals based on an averaged structural connectome for the classification of neuroimaging. *Artificial Intelligence in Medicine*, 106, 101870.*

8 - Mediation analysis issues. While I am not an expert in mediation analysis, I am doubtful about the interpretation of the results obtained. Can we really talk about causality in mediation analysis ? Maybe a slightly more convincing analysis method would be to use Bayesian Networks ? In addition,

how do the authors deal with multiple comparisons, when considering the large amount of tests that were performed ?

9 - I noted that the participants performed eyes closed resting state. is there anything that shows that participants indeed did not fall asleep ? Resting state is usually performed with eyes open.

10 - When using fMRIprep, users are strongly encouraged if not required to include the boilerplate text generated by fmripred, at least in supplementary material, and add all necessary citations. This also needs to include the precise version of fMRIprep used. Please adapt the text accordingly everywhere necessary.

11 - In the discussion (line 609), the following sentence is too much speculative, because it attempts at generalizing to not only unseen participants, but also other pathologies ! The authors may want to be more careful.

"We assume that the extent to which brain areas and their associated networks are affected in patients with neurodegenerative diseases, or in undiagnosed subjects with neurological symptoms, exceeds what would be expected among individuals undergoing normal aging. It follows then that, by way of example, if our results are shown to generalize for other neurological conditions, the diagnosis of a new patient might start by retrieving from his or her brain imaging data those GSP features that explain his or her serum NfL levels."

minor comments

12 - Line 906 It is not logical (line 906) to explain first the bootstrapping procedure for serum nfl levels , before explaining the PLSR setup (starts line 915)

13 - line 911 " ≥ 0.5 " I assume that means " > 0.5 "

14 - line 913 " ≥ 0.001 " here it should be " < 0.001 "

15 - In Supplementary Table 1 ? Line 215 : "we had 48 GSP features for HC subjects and 45 GSP features for ExPro subjects that had VIP scores greater than 1 (Suppl. Table 1)" and this sentence refers to the analysis of NfL levels. Therefore I infer that Supp. Table 1 contains VIP scores for all features, and the labels in parentheses correspond to groups : HC Healthy Controls and FA : Former Athletes ? Please add a caption to this figure and be consistent with the rest of the manuscript.

16 - Supplementary table 1 indicates that some features may explain Age at least as well as NfL. In paragraph "Serum NfL and Aging", the authors give some results regarding PLSR with age, but they don't specify which PLSR model is used ; is it the one with the limited set of GSP features after pruning ? Is it the full features set ?

--

Review by Nicolas Farrugia

Reviewer #2 (Remarks to the Author):

1. This review focuses on the PAI analyses.
2. The overall study is interesting and novel. The idea of correlating personality traits to distinct CNS pathology related to serum NfL for individuals with a history of mTBI and uninjured controls is a unique topic that provides value to the paper.
3. I would have liked to see the section on PAI expanded with more data and description of the samples' scores on PAI scales.
4. The references cited provide a rationale for examining the aggression scale in a TBI sample. It would improve understanding of the paper if this rationale was included in the text.
5. The methods section was positioned at the end of the manuscript, which was not what I expected and made the manuscript disjointed and difficult to follow. Specifically, before reading the PAI analyses I wanted to know more about the demographic and background information on the two groups, the sample sizes and details on the athletes' and controls' history of TBIs and other information, some of which was presented at the end in methods.
6. P. 17, lines 479-484: were all validity scales of the PAI within a normal range for all subjects? If any were elevated, were these cases excluded from the sample for analyses?
7. p. 17, lines 485-497: you state that aggression and depression scales differentiated HC and EXPRO groups and are related to TBI, but only describe subsequent analyses of the aggression scale. Provide an explanation for the lack of reporting depression scale results.
8. P. 17, line 491: PAI aggression raw scores were correlated with the low graph frequency feature from isthmus cingulate. What exactly was correlated with the PAI scores? That is, what scores were derived from the low graph frequency analysis of the isthmus cingulate that were used for this analysis? Further, what do these scores indicate? Are they a measure of correspondence between structural features and functional connections in this brain area? A higher number represents more correspondence? What distributional characteristics does this measure have for the 2 groups? Perhaps all or some of these questions are addressed in other sections of the manuscript, but in reviewing this part, the terminology has such a high level of abstraction that it is difficult to understand what this correlation might actually mean. If some translation or data could be presented in text or tables, it would be very helpful.
9. P. 17, lines 489 – 495: The low feature from the isthmus correlates with PAI aggression for healthy controls. But serum NfL does not correlate significantly with PAI aggression score for healthy controls. How are these results explained? Isn't the low feature from the isthmus the best predictor of NfL?
10. p. 24, line 791: reference here is Till et al, not Mill.

Reviewer #3 (Remarks to the Author):

This is an interesting paper representing a novel attempt to cross-correlate imaging and molecular biomarkers of axonal injury/axonal integrity. These findings are important, and cross-validate each of these biomarkers and represent an important step towards eventual clinical usefulness of these measures across a wide range of neurologic disorders. Potentially interesting relationships between

self-reported behavioral symptoms and cognitive functioning.

The imaging and biomarker assays are robust, and the analytical approaches appear state of the art. A reviewer with more biostatistical expertise should comment. The main limitation is the small sample size (n=20 healthy controls, and n=36 retired contact-sports athletes). Given the large number of analyses conducted and the expected co-linearity of the findings, it would be reassuring if these findings could be replicated in an independent cohort with imaging and biomarker data.

Reviewer #4 (Remarks to the Author):

The paper by Sihag and co authors, entitled “Functional Brain Signals Constrained by Structural Brain Connectivity Reveal Cohort Specific Features for Serum Neurofilament Light Chain” and submitted to the journal Communications Medicine, presents an investigation into linking Neurofilament-light measured via blood to the brain. The strengths of the paper include an interesting problem (linking NfL in the blood to brain structure and function in those with and without concussion history) and a novel computational approach. The weakness of the paper include the limited sample size, a lack of a clear reason to use the said computational approach over simpler and, perhaps, more interpretable methods, and a bit of a confusing presentation. Overall, while the approach used is novel, it is unclear why this analysis was chosen. Which when add to the vast number of comparisons in the paper, makes the paper confusing and going into too many directions at once. As such, more support of the paper is limited in its current form. I think streamlining the paper would vastly improve this work.

Why was the current analytical approach taken? For example, 132 features for only 20 subjects will certainly overfit. PLSR can help with that but it still seems like a non-optimal approach. So overall, I’m not sure I see the value of the explanatory analysis (i.e., non-cross-validated). Further, part of the result section tries to then project these PLSR results on to the brain, which is not be straightforward. Why not use a simple mass univariate approach to see which brain regions are associated with NfL? If nothing else, there would also allow the reader to see the advantage of the PLSR approach.

Are there any group differences? Either in the graph signal processing (GSP) features between the groups or in the correlations of the GSP features and NfL. Right now results are presented as a different in significance rather than significant differences between the groups.

The term “predict” is a bit overload in the paper. For example, both cross-validation and traditional explanatory approaches using predict or predictor. It would be helpful to change these terms in a way to distinguish the two analysis approaches. I am in favor of using predict only for the cross-validation and changing predictor to independent variable or something similar.

In general, there are a lot of analysis and it is not always clear how they relate or why they were performed. I think simplifying and streamlining the paper would help. Similar to the PLSR questions from above, why we these all performed? At points, the analysis seems to be complex for the sake of being complex. If the complexity is needed to answer the overarching question of the paper is not

clear.

Reviewer 1

1. A general feedback on the analytical approach of the paper. In its current form, the PLSR may appear insufficient for an expert in statistical learning, while being confusing for a reader that is not very knowledgeable on multivariate methods. First, for anyone familiar with machine learning, it is very surprising to explicitly present a set of analysis that extremely overfit on the training set, to then explain how to select features that don't overfit on the test set. The overall sample size is also extremely limited to perform PLSR. Some of the next questions in my review address alternative solutions to avoid this caveat. In the current manuscript, after the proposed PLSR analysis, the following and largest part of the analysis is done using more classical inference-based methods, using correlations and mediation analysis (which are more suited for this sample size). As a result, when trying to see the bigger picture in the approach taken, I have the impression that PLSR is solely performed in order to select a small set of relevant GSP features. PLSR was done using cross validation in order to enhance the predictive power, but in the rest of the analysis, inference-based statistics are used with bootstrapping. The mix of a prediction based and inference-based methods can be difficult to interpret, and might be considered a bit circular in the opinion of some readers and some experts in this community (see for example the series of papers by Danilo Bzdok on prediction versus inference). As a consequence, I am wondering what the authors think about the articulation of the different analytical approach used in the paper. For instance, how much of the analysis was hypothesis based versus data driven?
- A. We thank the reviewer for the feedback on the PLSR analysis and bringing to our attention the works of Danilo Bzdok on prediction vs inference in biostatistics (Bzdok et al., 2020) and neuroimaging (Bzdok, 2017). We address the points raised in this comment as follows.

Hypothesis vs Data-driven analyses: We hypothesized the GSP features to be associated with serum NfL due to its association with structural degeneration. However, there is a lack of neuroimaging studies that investigate the association of the serum NfL levels with specific brain regions even in healthy controls. In this context, our preliminary statistical analysis on GSP features from the whole brain was primarily data driven, leading to selection of a subset of features with high predictive relevance for serum NfL in the HC and ExPro cohorts. Subsequently, we hypothesized that aging was a causal factor for variation in serum NfL level and explored whether the GSP features mediated this relationship. In this context, there is no precedence in the literature, to the best of our knowledge, that establishes the associations between function-structure coupling in specific brain regions and age-driven variation in serum NfL levels. Also, the PLSR analysis did not provide any intuition into the role of GSP features in this context. Therefore, exploration of the mediating effects of GSP features on age-serum NfL relationship was data driven as well. The rest of the analyses focused on exploring associations between GSP features and the clinical and neurological observations

collected from the two cohorts, for which there is no established biological hypothesis to the best of our knowledge. The analyses in this context were data driven.

Prediction vs Inference: A mix of inference- and prediction-based analyses may bring up certain issues if used sequentially in a similar context as noted by the reviewer and in some recent works. However, we clarify that the inference-based analyses in our study are independent of the conclusions drawn from the prediction-based analyses. Specifically, the prediction performance of GSP features in the context of serum NfL did not influence their relevance to personality-based scores or structural markers such as cortical thickness and volumes, since the impact of observed serum NfL levels on these measures is itself not defined. Moreover, we did not use inference-based classical tests to validate the associations between serum NfL and GSP features isolated in the prediction-based analyses. In conclusion, we indeed chose to use only certain GSP features found to be relevant in the context of serum NfL for subsequent mediation and correlation analyses in order to keep the scope of the paper limited to serum NfL. However, since the association of the serum NfL levels with the investigated personality scores and structural measures is not known or leveraged, the problem of circular analysis or common issues associated with post variable selection inference as noted in (Bzdok, 2017) does not arise in our study.

From the works by Danilo Bzdok and reviewer's comments, we also recognized the distinct paradigms of inference and prediction-based statistics. To provide a complete picture to the reader and address any shortcomings of prediction-based analyses in selecting the appropriate features due to limited sample size, we have followed the reviewer's suggestion and augmented the analyses with a univariate variable selection procedure based on F-tests (see section on 'Serum NfL and GSP features' on page 11 and Results sections on page 13). In this analysis, we observed that several features deemed to be robust for prediction of NfL using cross validation in PLSR models in the HC and ExPro cohorts (see Fig. 3 in the revised manuscript) were also statistically significant in the F-test based variable selection procedure (see Fig. 2 in the revised manuscript), thus establishing the convergence between the two paradigms in our study.

PLSR Analysis: We adopted PLSR analysis in our study because of its recommended usage in the neuroimaging literature for scenarios with high multicollinearity among predictors and when the number of predictors outnumber the number of data samples (McIntosh et al., 2013, Lin et al., 2021). We recognize the concerns regarding the limitations, particularly overfitting, imposed by the data size. To address these, we performed rigorous non-parametric, permutation-based one-tailed tests on the explained variance (to investigate how likely it was to achieve a better performance from null distributions) to check for overfitting. In this context, we have reported the prediction performance of PLSR models with GSP features that did not show evidence of overfitting in the explanatory analysis. Irrespective of the limited data size, our experiments show evidence that the relevance of some GSP features (such as low graph frequency features from isthmus cingulate and posterior cingulate areas in the right

hemisphere) could be replicated in a larger cohort because of these features' predictive performance that translates across cohorts (see Supplementary Tables 4 and 5 and associated discussion).

2. How many components were used for PLSR? I have checked multiple times, and this was never mentioned in the paper, however this is a major defining feature in PLSR. The number of components will directly influence the capacity of the model to find a common space to link the inputs with the outputs. In addition, have the authors tested to vary the number of components?
 - A. We uniformly used one component for PLSR analysis in both cohorts when the features were selected after varying thresholds on VIP scores. The components of the PLSR models specific to HC and ExPro cohorts that resulted in the reported prediction performance based on leave-one-out cross validation are shown in Supplementary Appendices A and B in the Supplementary Material. Increasing the number of components in PLS models did not show any significant improvement in the performance.
3. Are the input and output features scaled or normalized?
 - A. The input and output features were z-score normalized for PLSR analysis.
4. Such questions would easily be solved if the authors could release the code for review; I have seen that the authors plan to release the code upon acceptance, but when it comes to such elaborated analysis, having the code for review would enable a much more efficient review process (at least for me!), as it would answer 90% of the questions I ask here.
 - A. We appreciate the reviewer's commitment to ensure the rigorousness of the results. To this end, we have included the code with the revised manuscript.
5. From Figure 2 it is really not obvious which ones are the "best" PLSR models. On line 240-241, the best model for ExPro is has 52% R2 but when looking at figure 2B, there are a few models around 0.5, and also models with higher R2 values, more than 0.6. Why aren't those last models better? In Figure 2A, however, the "best" model corresponds to the description in the text (58.6% R2) as there are only two significant models. Therefore, please clarify the choice of the "best" model for ExPro. This clarification is particularly important as it will decide which features are eventually kept for healthy subjects.
 - A. Thank you for the thoughtful comment. In the revised manuscript, we have modified the presentation of the PLSR analysis in the main manuscript to focus it on reporting the prediction performance. Therefore, we have selected only the GSP features that were observed to be more frequently present in the different prediction models (see Fig. 3 and associated discussion in the revised manuscript).

6. The overall methodology for selecting small sets of features (4 or 5 out of 132 GSP features!) using PLSR does not seem very sound to me considering the small number of examples. Because of the small sample size, the fact that filtering features with VIP scores larger than 1 still yields overfitting may indicate that the problem does not come from collinearity in features (as suggested by the authors), but from a low explained variance in many features. In addition, when setting very high thresholds in VIP, there are only 4 or 5 features left, which indicates that those are the 4 or 5 features that have a linear relationship with NfL scores in the first place. Could the authors justify better the PLSR approach performed here?
 - A. As noted in our response to point 1, PLSR is a recommended analysis approach in the neuroimaging literature for the scenarios with high multicollinearity among predictors and when the number of predictors outnumber the number of data samples (McIntosh et al., 2013, Lin et al., 2021). Since we did not have a biological hypothesis to select specific brain areas, we started with whole brain analysis, i.e., all 132 features and narrowed them down to a subset of features based on VIP scores, cross validation performance and tests for overfitting. We remark that the VIP based analysis in our manuscript is consistent with how it is performed in existing works as well (see Mehmood et al. 2012) and is augmented by non-parametric tests for overfitting. We have modified the statement regarding the overfitting by features with VIP score greater than 1 to include the reviewer's comment about the presence of large number of features with low explained variance in serum NfL (see text highlighted in red on Page 14 in the Supplementary Material).

7. Alternatively, because of all the drawbacks described in the previous comments, I suggest using a much simpler feature selection method, based on univariate statistics, such as F-tests (anova) directly with the output variables), or recursive feature elimination (which can also be done with cross validation), all that is explained here : https://scikit-learn.org/stable/modules/feature_selection.html#univariate-feature-selection . Notably, this feature selection method has been successfully applied to select relevant GSP features in previous work from my group, for classification (Menoret et al. 2017, Brahim & Farrugia 2020) and regression (Pilavci & Farrugia 2019, ICASSP). Such a method might be more convincing and easy to understand for the reader, as currently the PLSR setup seems a little convoluted, and critical details are missing (such as the number of components, the implementation used, ..see my earlier comments). Additionally, using univariate statistics for feature selection will be easier to articulate with the rest of the analysis performed, which are mostly based on inferential statistics.
 - A. Thank you for suggesting these alternative approaches for feature selection. We have augmented the analysis with a univariate statistics-based feature selection method that leverages F-tests (see section on 'Serum NfL and GSP features' on page 11 and Results sections on page 13). Also, since this line of analysis is driven by the strength of relationship between serum NfL and GSP features (quantified by F-scores and p-values),

we have discussed this analysis under the paradigm of inferential statistics and retained the prediction aspect of PLSR analysis under the paradigm of prediction-based statistics.

8. Mediation analysis issues. While I am not an expert in mediation analysis, I am doubtful about the interpretation of the results obtained. Can we really talk about causality in mediation analysis? Maybe a slightly more convincing analysis method would be to use Bayesian Networks? In addition, how do the authors deal with multiple comparisons, when considering the large amount of tests that were performed?

A. Thank you for the feedback on interpretation of mediation analysis. We concur that the scope of establishing causation from statistical tests that establish mediation is limited. In our analyses, we use the hypothesis that age is a causal factor for serum NfL levels which enabled us to select age as the independent variable and serum NfL as the dependent variable in the analysis. We have mentioned this explicitly in the revised manuscript (see discussion under 'GSP features complement and are independent of age in predicting serum NfL levels' on p. 17). In this context, we have modified our interpretations to the report whether GSP features mediate the age-serum NfL relationship. Exploring causality via Bayesian networks or other statistical approaches is an interesting direction. However, we believe that such an analysis must be pursued in conjunction with established biological causal hypotheses. We hope our study can trigger research in this direction and such analysis can be pursued in the future.

In this part of the analysis, we choose to investigate only the 9 GSP features found to be relevant in the context of serum NfL (determined through overlap in results in Fig. 2 and Fig. 3) for clinical and neurological interpretations. Therefore, the statistical significance of the results in Fig. 4 was assessed based on FDR correction of p-values obtained from 9 tests, each corresponding to a distinct GSP feature.

9. I noted that the participants performed eyes closed resting state. Is there anything that shows that participants indeed did not fall asleep? Resting state is usually performed with eyes open.

A. We clarify that the participants are spoken to between each sequence so at the beginning of the rs-fMRI and asked if OK to continue. The technicians did not proceed if the participant doesn't answer. We have also added this clarification on Page 8 of the revised manuscript under the subsection on 'Functional magnetic resonance imaging acquisition and processing'.

10. When using fMRIprep, users are strongly encouraged if not required to include the boilerplate text generated by fmriprep, at least in supplementary material, and add all necessary citations. This also needs to include the precise version of fMRIprep used. Please adapt the text accordingly everywhere necessary.

- A. Thank you for this suggestion. We have added the following text to the Supplementary Material.

Results included in this manuscript come from preprocessing performed using FMRIPREP version stable [1, 2, RRID:SCR_016216], a Nipype [3, 4, RRID:SCR_002502] based tool. Each T1w (T1-weighted) volume was corrected for INU (intensity non-uniformity) using N4BiasFieldCorrection v2.1.0 [5] and skull-stripped using antsBrainExtraction.sh v2.1.0 (using the OASIS template). Brain surfaces were reconstructed using recon-all from FreeSurfer v6.0.1 [6, RRID:SCR_001847], and the brain mask estimated previously was refined with a custom variation of the method to reconcile ANTs-derived and FreeSurfer-derived segmentations of the cortical gray-matter of Mindboggle [21, RRID:SCR_002438]. Spatial normalization to the ICBM 152 Nonlinear Asymmetrical template version 2009c [7, RRID:SCR_008796] was performed through nonlinear registration with the antsRegistration tool of ANTs v2.1.0 [8, RRID:SCR_004757], using brain-extracted versions of both T1w volume and template. Brain tissue segmentation of cerebrospinal fluid (CSF), white-matter (WM) and gray-matter (GM) was performed on the brain-extracted T1w using fast [17] (FSL v5.0.9, RRID:SCR_002823). Functional data was motion corrected using mcflirt (FSL v5.0.9 [9]). "Fieldmap-less" distortion correction was performed by co-registering the functional image to the same-subject T1w image with intensity inverted [13,14] constrained with an average fieldmap template [15], implemented with antsRegistration (ANTs). This was followed by co-registration to the corresponding T1w using boundary-based registration [16] with six degrees of freedom, using bregister (FreeSurfer v6.0.1). Motion correcting transformations, field distortion correcting warp, BOLD-to-T1w transformation and T1w-to-template (MNI) warp were concatenated and applied in a single step using antsApplyTransforms (ANTs v2.1.0) using Lanczos interpolation.

Physiological noise regressors were extracted applying CompCor [18]. Principal components were estimated for the two CompCor variants: temporal (tCompCor) and anatomical (aCompCor). A mask to exclude signal with cortical origin was obtained by eroding the brain mask, ensuring it only contained subcortical structures. Six tCompCor components were then calculated including only the top 5% variable voxels within that subcortical mask. For aCompCor, six components were calculated within the intersection of the subcortical mask and the union of CSF and WM masks calculated in T1w space, after their projection to the native space of each functional run. Frame-wise displacement [19] was calculated for each functional run using the implementation of Nipype. ICA-based Automatic Removal Of Motion Artifacts (AROMA) was used to generate aggressive noise regressors as well as to create a variant of data that is non-aggressively denoised [20].

Many internal operations of FMRIPREP use Nilearn [22, RRID:SCR_001362], principally within the BOLD-processing workflow. For more details of the pipeline see <https://fmriprep.readthedocs.io/en/stable/workflows.html>.

1. Esteban O, Markiewicz CJ, Blair RW, Moodie CA, Isik AI, Erramuzpe A, Kent JD, Goncalves M, DuPre E, Snyder M, Oya H, Ghosh SS, Wright J, Durnez J, Poldrack RA, Gorgolewski KJ. *fMRIPrep: a robust preprocessing pipeline for functional MRI*. *Nat Meth*. 2018; doi:[10.1038/s41592-018-0235-4](https://doi.org/10.1038/s41592-018-0235-4)
2. *fMRIPrep Available from: [10.5281/zenodo.852659](https://doi.org/10.5281/zenodo.852659)*.
3. Gorgolewski K, Burns CD, Madison C, Clark D, Halchenko YO, Waskom ML, Ghosh SS. *Nipype: a flexible, lightweight and extensible neuroimaging data processing framework in python*. *Front Neuroinform*. 2011 Aug 22;5(August):13. doi:[10.3389/fninf.2011.00013](https://doi.org/10.3389/fninf.2011.00013).
4. Gorgolewski KJ, Esteban O, Ellis DG, Nottter MP, Ziegler E, Johnson H, Hamalainen C, Yvernault B, Burns C, Manhães-Savio A, Jarecka D, Markiewicz CJ, Salo T, Clark D, Waskom M, Wong J, Modat M, Dewey BE, Clark MG, Dayan M, Loney F, Madison C, Gramfort A, Keshavan A, Berleant S, Pinsard B, Goncalves M, Clark D, Cipollini B, Varoquaux G, Wassermann D, Rokem A, Halchenko YO, Forbes J, Moloney B, Malone IB, Hanke M, Mordom D, Buchanan C, Pauli WM, Huntenburg JM, Horea C, Schwartz Y, Tungaraza R, Iqbal S, Kleesiek J, Sikka S, Frohlich C, Kent J, Perez-Guevara M, Watanabe A, Welch D, Cumba C, Ginsburg D, Eshaghi A, Kastman E, Bougacha S, Blair R, Acland B, Gillman A, Schaefer A, Nichols BN, Giavasis S, Erickson D, Correa C, Ghayoor A, Küttner R, Haselgrove C, Zhou D, Craddock RC, Haehn D, Lampe L, Millman J, Lai J, Renfro M, Liu S, Stadler J, Glatard T, Kahn AE, Kong X-Z, Triplett W, Park A, McDermottroe C, Hallquist M, Poldrack R, Perkins LN, Noel M, Gerhard S, Salvatore J, Mertz F, Broderick W, Inati S, Hinds O, Brett M, Durnez J, Tambini A, Rothmei S, Andberg SK, Cooper G, Marina A, Mattfeld A, Urchs S, Sharp P, Matsubara K, Geisler D, Cheung B, Floren A, Nickson T, Pannetier N, Weinstein A, Dubois M, Arias J, Tarbert C, Schlamp K, Jordan K, Liem F, Saase V, Harms R, Khanuja R, Podranski K, Flandin G, Papadopoulos Orfanos D, Schwabacher I, McNamee D, Falkiewicz M, Pellman J, Linkersdörfer J, Varada J, Pérez-García F, Davison A, Shachnev D, Ghosh S. *Nipype: a flexible, lightweight and extensible neuroimaging data processing framework in Python*. 2017. doi:[10.5281/zenodo.581704](https://doi.org/10.5281/zenodo.581704).
5. Tustison NJ, Avants BB, Cook PA, Zheng Y, Egan A, Yushkevich PA, Gee JC. *N4ITK: improved N3 bias correction*. *IEEE Trans Med Imaging*. 2010 Jun;29(6):1310–20. doi:[10.1109/TMI.2010.2046908](https://doi.org/10.1109/TMI.2010.2046908).
6. Dale A, Fischl B, Sereno MI. *Cortical Surface-Based Analysis: I. Segmentation and Surface Reconstruction*. *Neuroimage*. 1999;9(2):179–94. doi:[10.1006/nimg.1998.0395](https://doi.org/10.1006/nimg.1998.0395).
7. Fonov VS, Evans AC, McKinstry RC, Almlí CR, Collins DL. *Unbiased nonlinear average age-appropriate brain templates from birth to adulthood*. *NeuroImage*; Amsterdam. 2009 Jul 1;47:S102. doi:[10.1016/S1053-8119\(09\)70884-5](https://doi.org/10.1016/S1053-8119(09)70884-5).
8. Avants BB, Epstein CL, Grossman M, Gee JC. *Symmetric diffeomorphic image registration with cross-correlation: evaluating automated labeling of elderly and*

neurodegenerative brain. *Med Image Anal.* 2008 Feb;12(1):26–41.
doi:[10.1016/j.media.2007.06.004](https://doi.org/10.1016/j.media.2007.06.004).

9. Jenkinson M, Bannister P, Brady M, Smith S. Improved optimization for the robust and accurate linear registration and motion correction of brain images. *Neuroimage.* 2002 Oct;17(2):825–41. doi:[10.1006/nimg.2002.1132](https://doi.org/10.1006/nimg.2002.1132).

10. Andersson JLR, Skare S, Ashburner J. How to correct susceptibility distortions in spin-echo echo-planar images: application to diffusion tensor imaging. *Neuroimage.* 2003 Oct;20(2):870–88. doi:[10.1016/S1053-8119\(03\)00336-7](https://doi.org/10.1016/S1053-8119(03)00336-7).

11. Cox RW. AFNI: software for analysis and visualization of functional magnetic resonance neuroimages. *Comput Biomed Res.* 1996 Jun;29(3):162–73.
doi:[10.1006/cbmr.1996.0014](https://doi.org/10.1006/cbmr.1996.0014).

12. Jenkinson M. Fast, automated, N-dimensional phase-unwrapping algorithm. *Magn Reson Med.* 2003 Jan;49(1):193–7. doi:[10.1002/mrm.10354](https://doi.org/10.1002/mrm.10354).

13. Huntenburg JM. Evaluating nonlinear coregistration of BOLD EPI and T1w images. Freie Universität Berlin; 2014. Available from: <http://hdl.handle.net/11858/00-001M-0000-002B-1CB5-A>.

14. Wang S, Peterson DJ, Gatenby JC, Li W, Grabowski TJ, Madhyastha TM. Evaluation of Field Map and Nonlinear Registration Methods for Correction of Susceptibility Artifacts in Diffusion MRI. *Front Neuroinform.* 2017 [cited 2017 Feb 21];11.
doi:[10.3389/fninf.2017.00017](https://doi.org/10.3389/fninf.2017.00017).

15. Treiber JM, White NS, Steed TC, Bartsch H, Holland D, Farid N, McDonald CR, Carter BS, Dale AM, Chen CC. Characterization and Correction of Geometric Distortions in 814 Diffusion Weighted Images. *PLoS One.* 2016 Mar 30;11(3):e0152472.
doi:[10.1371/journal.pone.0152472](https://doi.org/10.1371/journal.pone.0152472).

16. Greve DN, Fischl B. Accurate and robust brain image alignment using boundary-based registration. *Neuroimage.* 2009 Oct;48(1):63–72.
doi:[10.1016/j.neuroimage.2009.06.060](https://doi.org/10.1016/j.neuroimage.2009.06.060).

17. Zhang Y, Brady M, Smith S. Segmentation of brain MR images through a hidden Markov random field model and the expectation-maximization algorithm. *IEEE Trans Med Imaging.* 2001 Jan;20(1):45–57. doi:[10.1109/42.906424](https://doi.org/10.1109/42.906424).

18. Behzadi Y, Restom K, Liu J, Liu TT. A component based noise correction method (CompCor) for BOLD and perfusion based fMRI. *Neuroimage.* 2007 Aug 1;37(1):90–101.
doi:[10.1016/j.neuroimage.2007.04.042](https://doi.org/10.1016/j.neuroimage.2007.04.042).

19. Power JD, Mitra A, Laumann TO, Snyder AZ, Schlaggar BL, Petersen SE. Methods to detect, characterize, and remove motion artifact in resting state fMRI. *Neuroimage*. 2013 Aug 29;84:320–41. doi:[10.1016/j.neuroimage.2013.08.048](https://doi.org/10.1016/j.neuroimage.2013.08.048).

20. Pruim RHR, Mennes M, van Rooij D, Llera A, Buitelaar JK, Beckmann CF. ICA-AROMA: A robust ICA-based strategy for removing motion artifacts from fMRI data. *Neuroimage*. 2015 May 15;112:267–77. doi:[10.1016/j.neuroimage.2015.02.064](https://doi.org/10.1016/j.neuroimage.2015.02.064).

21. Klein A, Ghosh SS, Bao FS, Giard J, Häme Y, Stavsky E, et al. Mindboggling morphometry of human brains. *PLoS Comput Biol* 13(2): e1005350. 2017. doi:[10.1371/journal.pcbi.1005350](https://doi.org/10.1371/journal.pcbi.1005350).

22. Abraham A, Pedregosa F, Eickenberg M, Gervais P, Mueller A, Kossaifi J, Gramfort A, Thirion B, Varoquaux G. Machine learning for neuroimaging with scikit-learn. *Front in Neuroinf* 8:14. 2014. doi:[10.3389/fninf.2014.00014](https://doi.org/10.3389/fninf.2014.00014).

11. In the discussion (line 609), the following sentence is too much speculative, because it attempts at generalizing to not only unseen participants, but also other pathologies! The authors may want to be more careful.

"We assume that the extent to which brain areas and their associated networks are affected in patients with neurodegenerative diseases, or in undiagnosed subjects with neurological symptoms, exceeds what would be expected among individuals undergoing normal aging. It follows then that, by way of example, if our results are shown to generalize for other neurological conditions, the diagnosis of a new patient might start by retrieving from his or her brain imaging data those GSP features that explain his or her serum NfL levels."

- A. Thank you for this thoughtful comment. We have modified the discussion to the following: 'It follows then that, by way of example, if our results are shown to generalize for other cohorts, the diagnosis of neurodegeneration in a new patient might be facilitated by identifying those GSP features obtained from their neuroimaging data that are most strongly associated with their serum NfL levels and then determining if these features constitute a normal age-associated correlation or a pathological association.' (see p. 24, l. 750 of the revised manuscript).

12. Line 906 It is not logical (line 906) to explain first the bootstrapping procedure for serum NfL levels, before explaining the PLSR setup (starts line 915).

- A. Thank you for pointing this out. We have moved the discussion on the non-parametric test procedure after PLSR setup (see p. 11, l. 340).

13. line 911 " ≥ 0.5 " I assume that means " > 0.5 ", line 913 " ≥ 0.001 " here it should be " < 0.001 "

A. Thank you for catching these typos. We have made the necessary corrections.

14. In Supplementary Table 1 Line 215: "we had 48 GSP features for HC subjects and 45 GSP features for ExPro subjects that had VIP scores greater than 1 (Suppl. Table 1)" and this sentence refers to the analysis of NfL levels. Therefore, I infer that Supp. Table 1 contains VIP scores for all features, and the labels in parentheses correspond to groups: HC Healthy Controls and FA: Former Athletes? Please add a caption to this figure and be consistent with the rest of the manuscript.

A. Thank you for catching this inconsistency. We have fixed the labels for this table (Supp. Table 8 in supplementary material) and added a caption.

15. Supplementary table 1 indicates that some features may explain Age at least as well as NfL. In paragraph "Serum NfL and Aging", the authors give some results regarding PLSR with age, but they don't specify which PLSR model is used; is it the one with the limited set of GSP features after pruning? Is it the full features set?

A. In this paragraph, we discussed only the results of linear regression between age and serum NfL in the two cohorts. Therefore, there were no GSP features involved. However, since these results are tangential to the main focus of testing relevance of GSP features in context of serum NfL, we have moved them to section XI of the supplementary material in this revision.

Reviewer 2

1. I would have liked to see the section on PAI expanded with more data and description of the samples' scores on PAI scales.

A. Based on reviewer's feedback, we have added the details on the mean scores in the two cohorts in Table 1 in the revised manuscript.

2. The references cited provide a rationale for examining the aggression scale in a TBI sample. It would improve understanding of the paper if this rationale was included in the text.

A. Thank you for this suggestion. We have included the rationale that the examined scales are considered valid for TBI as these were known to be not confounded by transdiagnostic measures characteristic of both psychopathology and neuropathology (see highlighted text on Page 7 under PAI assessments subsection).

3. The methods section was positioned at the end of the manuscript, which was not what I expected and made the manuscript disjointed and difficult to follow. Specifically, before reading the PAI analyses I wanted to know more about the demographic and background information on the two groups, the sample sizes and details on the athletes'

and controls' history of TBIs and other information, some of which was presented at the end in methods.

- A. Based on this feedback, we have restructured the manuscript. Mainly, we have moved the methods section (renamed to 'Methods and Materials, Page 6) before the results section. In the revised methods section, we have organized the presentation into following subsections:
- a) Participants (Page 6):
 - Demographic information
 - PAI Assessment description
 - Neuroimaging data and serum NfL acquisition and processing
 - b) Data analysis (Page 9):
 - Graph signal processing-based feature extraction
 - Serum NfL and GSP features
 - ◆ Association between GSP features and serum NfL
 - ◆ GSP features as predictors of serum NfL
 - *Nonparametric permutation test for PLSR*
 - *Variable selection for prediction model*
 - *Prediction performance based on cross validation*
 - c) Clinical and neurological interpretations of GSP features linked with serum NfL (Page 12)
4. P. 17, lines 479-484: were all validity scales of the PAI within a normal range for all subjects? If any were elevated, were these cases excluded from the sample for analyses?
- A. We clarify that all validity scales of the PAI were within a normal range for these subjects. Any subject not in the normal range on the validity scales was excluded from the study.
5. p. 17, lines 485-497: you state that aggression and depression scales differentiated HC and EXPRO groups and are related to TBI, but only describe subsequent analyses of the aggression scale. Provide an explanation for the lack of reporting depression scale results.
- A. We did not find any statistically significant observations in the context of depression scale in our experiments. We have clarified this in the manuscript (see highlighted text under subsection 'GSP features in NfL prediction models are associated with personality scores in healthy controls' on Page 20).
6. P. 17, line 491: PAI aggression raw scores were correlated with the low graph frequency feature from isthmus cingulate. What exactly was correlated with the PAI scores? That

is, what scores were derived from the low graph frequency analysis of the isthmus cingulate that were used for this analysis? Further, what do these scores indicate? Are they a measure of correspondence between structural features and functional connections in this brain area? A higher number represents more correspondence? What distributional characteristics does this measure have for the 2 groups? Perhaps all or some of these questions are addressed in other sections of the manuscript, but in reviewing this part, the terminology has such a high level of abstraction that it is difficult to understand what this correlation might actually mean. If some translation or data could be presented in text or tables, it would be very helpful.

- A. Thank you for this feedback. Low graph frequency feature from isthmus cingulate indeed represents a measure of correspondence between structural and functional connectomes in this region, with a higher number representing more correspondence. Therefore, our analysis investigated the statistical correspondence between raw aggression scores derived from PAI and low graph frequency feature from isthmus cingulate. A positive correlation implies that higher aggression score is characterized by stronger structure-function alignment in the isthmus cingulate.

We have restructured the manuscript to include methods and materials before the results section. This helps in establishing the notion of graph filters. To illustrate the graph filtered outputs of BOLD data, we have included a figure in the supplementary file (Supplementary Figure 1) which further provides a pictorial representation of low and high graph frequency features. We have also referred to this figure at the start of the subsection on 'Clinical and neurological interpretations of GSP features' in the results section (see text highlighted in red on Page 17) to remind the readers about the significance of the low and high graph frequency features. The distributional differences between the two groups were investigated in our previous study. Since the focus of this paper is on serum NfL, we have included the differences in terms of predictive or inferential characteristics in the GSP features for the two groups in Supplementary Tables 1 and 2 in the supplementary document. From these results, we find that the low graph frequency feature from isthmus cingulate had more significant correlation with serum NfL in the HC cohort as compared to that in ExPro cohort (zscore = -2.08, p-value = 0.0368).

7. P. 17, lines 489 – 495: The low feature from the isthmus correlates with PAI aggression for healthy controls. But serum NfL does not correlate significantly with PAI aggression score for healthy controls. How are these results explained? Isn't the low feature from the isthmus the best predictor of NfL?
- A. The low graph frequency feature from isthmus cingulate is correlated with both serum NfL (partial correlation = -0.5425) and PAI aggression score (correlation = 0.7052) in healthy controls. However, PAI aggression score was not significantly associated with serum NfL in HC cohort (correlation = -0.239 , p-value = 0.31) and therefore, was not

reported in the previous version of the manuscript. The smaller correlation between serum NfL and PAI aggression score in HC cohort as compared to ExPro cohort merits further neurological exploration, which was not the focus of our work. However, note that we also concurrently observed smaller correlation between the GSP feature from isthmus cingulate and PAI aggression score in ExPro cohort (correlation = 0.1873) in our experiments which corroborates the relevance of GSP features in understanding the brain mechanisms behind aggression and differentiation of serum NfL levels in the two populations. We have added this discussion in the revised manuscript (see green colored text on Page 20).

8. p. 24, line 791: reference here is Till et al, not Mill.
- A. Thanks for catching this error. We have corrected this.

Reviewer 3

1. This is an interesting paper representing a novel attempt to cross-correlate imaging and molecular biomarkers of axonal injury/axonal integrity. These findings are important, and cross-validate each of these biomarkers and represent an important step towards eventual clinical usefulness of these measures across a wide range of neurologic disorders. Potentially interesting relationships between self-reported behavioral symptoms and cognitive functioning.

The imaging and biomarker assays are robust, and the analytical approaches appear state of the art. A reviewer with more biostatistical expertise should comment. The main limitation is the small sample size (n=20 healthy controls, and n=36 retired contact-sports athletes). Given the large number of analyses conducted and the expected co-linearity of the findings, it would be reassuring if these findings could be replicated in an independent cohort with imaging and biomarker data.

- A. Thank you for the positive feedback. We agree that replicating our findings in an independent cohort would be ideal. However, due to logistical constraints, obtaining an independent dataset with neuroimaging and clinical data in a reasonable time is infeasible.

We remark that we have used rigorous statistics, including bootstrapping and non-parametric, permutation-based tests to test the statistical significance of our findings wherever applicable. Also, despite significant neurological heterogeneity in the HC and ExPro cohorts, our experiments show that prediction performance of GSP features from isthmus cingulate and posterior cingulate in the right hemisphere in context of serum NfL was transferable across the cohorts (see Supp. Table 4 and Supp. Table 5 in the supplementary material).

Reviewer 4

1. Why was the current analytical approach taken? For example, 132 features for only 20 subjects will certainly overfit. PLSR can help with that but it still seems like a non-optimal approach. So overall, I'm not sure I see the value of the explanatory analysis (i.e., non-cross-validated). Further, part of the result section tries to then project these PLSR results on to the brain, which is not straightforward. Why not use a simple mass univariate approach to see which brain regions are associated with NfL? If nothing else, there would also allow the reader to see the advantage of the PLSR approach.

A. We adopted PLSR analysis in our study because of its recommended usage in the neuroimaging literature for the scenarios with high multicollinearity among predictors and when the number of predictors outnumber the number of data samples (McIntosh et al., 2013, Lin et al., 2021). Our goal was to test the relevance of serum NfL features in context of serum NfL for HC and ExPro cohorts and PLSR analysis aided in this attempt. The role of explanatory analysis was to report the best explanatory performance without overfitting. However, in the revised manuscript, we have focused the presentation of PLS models only on the prediction performance. In this context, we have added Figure 3 to depict the most robust predictors after the cross validation procedure in each cohort. Furthermore, we have moved the explanatory analysis to the Supplementary Material (see section VII in the Supplementary Material) to avoid confusion for the readers.

The PLSR models consisted of only one component and therefore, we could associate distinct weights with the selected brain areas. Following reviewer's suggestion, we adopted an F-test based univariate feature selection approach (also recommended by Reviewer 1). The results are reported in Fig. 2 with accompanying discussions on setup in the section 'Serum NfL and GSP features' on page 11 and results in the Results sections on page 13. We also observed convergence in the results of PLSR and univariate feature selection approach in terms of the GSP features deemed relevant for serum NfL in both cohorts.

Both prediction-based and inference paradigms of statistics can provide distinct insights into the relationships in the data (Bzdok et al., 2020). Therefore, we have retained the PLSR analysis as a prediction-based statistical approach while discussing the univariate approach under inferential statistics paradigm. Prediction based analysis provides insight into the performance of the statistical model on unseen data which offers advantages in understanding the applicability of the results in clinical settings. For instance, higher serum NfL levels are often reported in various neurodegenerative contexts, both before and after the onset of disease, with no underlying statistical model that explains the increase with respect to a healthy population. In this context, the use of predictive models may aid in prognosis in asymptomatic individuals if

divergence is observed in predicted serum NfL and observed serum NfL over time. Such an application merits further research and corroboration of our findings on other cohorts, which is better aided by reporting of predictive performances than statistical associations.

2. Are there any group differences? Either in the graph signal processing (GSP) features between the groups or in the correlations of the GSP features and NfL. Right now results are presented as a difference in significance rather than significant differences between the groups.
 - A. The group differences in GSP features were investigated in our previous work (Sihag et al., 2020). In Supplementary Table 1, we have reported the differences between correlations of GSP features and serum NfL in the two cohorts. In this context, we observe differences in association of the three low graph frequency features from isthmus cingulate and parahippocampus areas in the right hemisphere (both were relevant for serum NfL in HC cohorts) and high graph frequency features from transverse temporal and lingual areas in the left hemisphere (both were relevant for serum NfL in ExPro cohorts).
3. The term “predict” is a bit overload in the paper. For example, both cross-validation and traditional explanatory approaches using predict or predictor. It would be helpful to change these terms in a way to distinguish the two analysis approaches. I am in favor of using predict only for the cross-validation and changing predictor to independent variable or something similar.
 - A. Thank you for this suggestion. We have modified the discussions to refer to prediction only in the context of cross-validation performance. Furthermore, we have modified the discussion on PLSR analysis to imply that the prediction performance from cross-validation is the primary measure of interest in this analysis (see discussion under ‘Prediction performance based on cross-validation’ on page 12). We have also moved the explanatory analysis to Section VII in the Supplementary Material to avoid confusion.
4. In general, there are a lot of analysis, and it is not always clear how they relate or why they were performed. I think simplifying and streamlining the paper would help. Similar to the PLSR questions from above, why we these all performed? At points, the analysis seems to be complex for the sake of being complex. If the complexity is needed to answer the overarching question of the paper is not clear.
 - A. We thank the reviewer for this feedback on presentation of results in the paper. We have made major changes to the organization of methods and results sections and moved some results to the supplementary file. In the revised methods section, we have organized the discussion into following subsections:
 - a) Participants (Page 6):

- Demographic information
 - PAI Assessment description
 - Neuroimaging data and serum NfL acquisition and processing
- b) Data analysis (Page 9):
- Graph signal processing-based feature extraction
 - Serum NfL and GSP features
 - ◆ Association between GSP features and serum NfL
 - ◆ GSP features as predictors of serum NfL
 - *Nonparametric permutation test for PLSR*
 - *Variable selection for prediction model*
 - *Prediction performance based on cross validation*
- c) Clinical and neurological interpretations of GSP features linked with serum NfL (Page 12)

We moved the methods section before the results section to provide a better understanding into the dataset and the problem being investigated in the paper. Furthermore, the methods section has been modified to include a specific data analysis sub-section divided into three parts. In the first part, we describe the relevant theoretical tools from graph signal processing that enable us to characterize the measures of function-structure coupling extracted from BOLD activity in different brain areas. Next, we discuss the primary aim of the paper, which is to investigate the relevance of these measures in the context of serum NfL in the two cohorts. For this purpose, we have organized the statistical approaches under inference based and prediction-based paradigms. Inference and prediction-based paradigms provide distinct insights into the statistical relationships between GSP features and serum NfL (Bzdok et al., 2020). We describe the univariate feature selection approach and PLSR based prediction analyses in this sub-section. Finally, we discuss our results in the context of clinical and neurological interpretations of the GSP features found to be relevant for serum NfL, where we observe that aggression related personality assessment scores and structural measures correlate with these GSP features.

References:

1. Bzdok, Danilo. "Classical statistics and statistical learning in imaging neuroscience." *Frontiers in neuroscience* 11 (2017): 543.
2. Bzdok, Danilo, Denis Engemann, and Bertrand Thirion. "Inference and Prediction Diverge in Biomedicine." *Patterns* 1.8 (2020): 100119.
3. McIntosh, Anthony R., and Bratislav Mišić. "Multivariate statistical analyses for neuroimaging data." *Annual review of psychology* 64 (2013): 499-525.
4. Lin, Fa-Hsuan, Hsin-Ju Lee, Wen-Jui Kuo, and Iiro P. Jääskeläinen. "Multivariate Identification of Functional Neural Networks Underpinning Humorous Movie Viewing." *Frontiers in psychology* 11 (2021): 4008.
5. Sihag, S., S. Naze, F. Taghdiri, C. Tator, R. Wennberg, D. Mikulis, R. Green, B. Colella, M. C. Tartaglia, and J. R. Kozloski. 2020. "Multimodal Dynamic Brain Connectivity Analysis Based on Graph Signal Processing for Former Athletes with History of Multiple Concussions." *IEEE Transactions on Signal and Information Processing over Networks* 6: 284–99.

Reviewers' comments:

Reviewer #1 (Remarks to the Author):

Thanks a lot for this much improved version. The choices made for the current version have strengthened the paper significantly.

(1) The addition of the univariate feature selection method is very welcome, and the results are quite consistent with PLSR. I think the overall readability and clarity could be improved because currently the articulation between the two types of analysis (PLSR and univ.) feels a little artificial. First, the authors did include a paragraph explaining the difference between prediction and inference, but this paragraph would be better placed at the beginning of the section ; this would enable to do an introduction of the overall analytical framework, then introduce univariate FS, then introduce PLSR. There is a similar problem in the results section; the two types of analysis are presented in two different subsections as two different set of results; but the paper would gain in clarity in they were presented together. Actually, in the figure 3 caption the authors refer to 'robustness' of features, which is a good angle to present those cross validated results. However the results with Figure 2 are quite similar, when comparing Figures 2 and 3. Overall, I think the articulation between the two types of analysis can be improved for clarity, at all stages in the paper : methods, results and discussion.

(2) Something I did not mention in my first review: the associations between GSP features and structural brain features are interesting, but as they are presented as "uncorrected", it raises the question whether other brain structural features in cortical thickness or volume would also show up as significant at an uncorrected level. In addition, the two selected features (pericalcarine thickness and amygdala volume) are not mentioned in the introduction of the paper as hypothesis. It would be wiser to have explicit hypothesis on those areas to justify their choice, rather than reporting them directly in the results section with no justification and at an uncorrected statistical threshold level. (section III C 3)

(3) The new appendix A and B in the supplementary material are a very nice addition. It shows how feature selection varies in PLSR according to the subjects. It s a good idea to show all subjects, but maybe a plot with all subjects together would be more informative ; something like a 2D image of weights for all features selected across all subjects, with subjects on the y axis and features on the x axis, for example ? This would better show the commonalities across subjects, and would be a nicer summary for the reader.

(4) Thanks for adding the code and features data. This should definitely be uploaded on a public website and linked properly in the main text ; this consitute a very valuable addition. Some minor comments on the python code ; (a) the jupyter notebook with the univariate feature selection contains python imports which are not used. (b) It would be good to have a few more comments explaining what is done in the cells (c) as well as a very basic visualisation of the results, by plotting the p values (or the negative log p values) with the ROI labels, of such kind of visualization.

Reviewer #2 (Remarks to the Author):

The revision of this paper is much improved. The re-organization and additional explanation make it easier to read and understand. The authors have addressed all comments related to the section on the PAI. I have no further comments and recommend this version of the manuscript for publication.

Reviewer #4 (Remarks to the Author):

The authors have done a reasonable job answering my previous questions. I have one question/comment that still remains. It would be interesting to discuss the authors graph signal processing which relies on having DTI data as well as fMRI data in comparison to newer approaches that do not rely on DTI data. See for example:

Gao S, Xia X, Scheinost D, Mishne G. Smooth graph learning for functional connectivity estimation. *Neuroimage*. 2021 Oct 1;239:118289. doi: 10.1016/j.neuroimage.2021.118289. Epub 2021 Jun 23. PMID: 34171497.

Reviewer 1

1. A The addition of the univariate feature selection method is very welcome, and the results are quite consistent with PLSR. I think the overall readability and clarity could be improved because currently the articulation between the two types of analysis (PLSR and univ.) feels a little artificial. First, the authors did include a paragraph explaining the difference between prediction and inference, but this paragraph would be better placed at the beginning of the section; this would enable to do an introduction of the overall analytical framework, then introduce univariate FS, then introduce PLSR. There is a similar problem in the results section; the two types of analysis are presented in two different subsections as two different set of results; but the paper would gain in clarity in they were presented together. Actually, in the figure 3 caption the authors refer to 'robustness' of features, which is a good angle to present those cross validated results. However the results with Figure 2 are quite similar, when comparing Figures 2 and 3. Overall, I think the articulation between the two types of analysis can be improved for clarity, at all stages in the paper: methods, results and discussion.

- A. Thank you for the thoughtful comments. We have made the suggested changes to the discussions and presentations of the two types of analysis. In Section II.B.2, we have moved the discussion on prediction vs inference to the beginning and modified the previous discussion to the following (page 11):

Prediction and inference form the two paradigms of statistical analysis that provide distinct insights into the relevance of variables depending on the actual modeling goal (Bzdok et al., 2019). Inference helps in isolating individual variables that are significantly associated with the target variable (in this case, serum NfL) whereas prediction driven analysis guides the isolation of variables deemed relevant for predicting the target variable in unseen data. In this study, we aimed to explore the statistical correspondence of GSP features in the context of serum NfL levels in the two cohorts under both statistical paradigms. Due to lack of neuroimaging studies that link specific brain regions with serum NfL, we adopted data-driven approaches to isolate the GSP features that were most relevant to serum NfL from the complete set of 132 features.

We have merged the subsections in the results section to form one section (Section III.A) that discusses both prediction and inference paradigms. We have also edited this section to smoothen the transition between the discussions on both sets of results (edits are marked in red). We have also augmented Figure 3 with carpet plots (Fig. 3A and Fig. 3D) that show the distribution of the selection of most robust features across individual subjects in the two cohorts. These new plots are similar to what was suggested in point 3 below. Discussions on the revised plots are added in Section III.A (green colored text).

Lastly, in the Results section, we have added the following discussion at the beginning.

The current study and results add a new dimension to the analysis of serum NfL levels in the context of traumatic brain injury and, by association, neurodegenerative diseases by demonstrating that brain activity patterns decomposed over the brain's structure are partially explanatory and predictive of serum NfL levels in two distinct cohorts. For each cohort, we observed a convergence between the findings from inference and predictive paradigms of the statistical analyses. Our results showed that the low graph frequency feature from isthmus cingulate in the right hemisphere (which is positively correlated with structure-function coupling in this area) had the strongest association and most robust predictive performance for serum NfL in healthy controls. In contrast, the low graph frequency features from superior frontal and caudal anterior cingulate in the right hemisphere and high graph frequency features from temporal lobe and lingual in the left hemisphere (which are negatively correlated with structure-function coupling in the corresponding areas) had both the strongest association and predictive relevance for serum NfL in the ExPro cohort. Therefore, our analyses clearly established the significance and heterogeneity of our neuroimaging biomarkers associated with serum NfL in two different populations with similar serum NfL levels.

2. Something I did not mention in my first review: the associations between GSP features and structural brain features are interesting, but as they are presented as "uncorrected", it raises the question whether other brain structural features in cortical thickness or volume would also show up as significant at an uncorrected level. In addition, the two selected features (pericalcarine thickness and amygdala volume) are not mentioned in the introduction of the paper as hypothesis. It would be wiser to have explicit hypothesis on those areas to justify their choice, rather than reporting them directly in the results section with no justification and at an uncorrected statistical threshold level. (section III C 3)
 - A. Thank you for this feedback. The statistically significant differences in aggression and depression related personality assessment scores prompted us to investigate the relevance of amygdala volume in the context of GSP features and serum NfL. Cortical thinning is a marker of white matter degeneration and therefore, was investigated in the context of serum NfL. We have added the following discussion in the introduction to communicate these aspects to the reader.

To explore the clinical and neurological interpretation of the GSP features associated with serum NfL in our experiments, we also tested their associations with cognitive scores and structural measures. The two cohorts differed significantly in terms of aggression and depression related personality assessment scores (discussed in Section II.A.1). Therefore, it was of interest to explore the associations of GSP features with amygdala, since this region is instrumental in a broader neural circuit responsible for modulating aggression (Matthies et al., 2012 and Gouveia et al., 2019) and has been implicated in depression related disorders (Daftary et al., 2018, Hamilton et al., 2008, Kim et al., 2021). Furthermore, we hypothesized that the links between white matter

degeneration and serum NfL levels might also be characterized by reduced cortical thickness (Spotorno et al., 2020). In this context, we hypothesized that those GSP features aligned to our hypothesis of serum NfL being associated with conformity of BOLD activity to white matter anatomy might also be associated with cortical thickness measures. In a broader context, cortical thinning is also neurologically and clinically relevant, as it has been associated with structural abnormalities after TBI (Santhanam et al., 2019) as well as pathological personality traits (Sheehan et al., 2021).

We have further emphasized on analyzing the cortical thickness of pericalcarine region in the right hemisphere by including the following discussions on its association with serum NfL and neurological interpretations in Section III.B.3 (page 22-23).

There was no statistically significant difference in the cortical thickness of different regions in the two cohorts. We observed that for the HC cohort, the thickness of pericalcarine region in the right hemisphere was negatively correlated with serum NfL (partial correlation with correction for age, $\rho = -0.622$, p -value = 0.003), which was in line with our hypothesis that cortical thickness may be negatively correlated with serum NfL. The significant correlation after correction for age indicated that this association may not be driven by aging in HC cohort. We did not observe any other cortical thickness measures to be correlated with serum NfL for ExPro or HC subjects at 0.01 significance level (uncorrected). Interestingly, cortical thinning of pericalcarine region is linked with impulsive and risky tendencies in the existing studies (Miglin et al. 2019) which characterize the behavioral impacts of TBI. Therefore, we further investigated the association of the pericalcarine thickness with serum NfL and GSP features in both cohorts.

For amygdala, the volume in the left hemisphere was not observed to be relevant for serum NfL. Therefore, we focused our experiments only on its volume in the right hemisphere. We have clarified this with the following discussion in Section III.B.3.

Volume of amygdala in the left hemisphere was significantly associated with age ($\rho = -0.3926$, p -value = 0.0179) for ExPro subjects but not for HC subjects. No significant associations with serum NfL were observed for volume of amygdala in the left hemisphere for both either cohorts. We hypothesized that aging was the driving factor behind the change in volume of amygdala in both hemispheres in ExPro subjects. We focused our subsequent analysis only the volume of amygdala in the right hemisphere since it was observed to be relevant for serum NfL in ExPro subjects.

3. The new appendix A and B in the supplementary material are a very nice addition. It shows how feature selection varies in PLSR according to the subjects. It's a good idea to show all subjects, but maybe a plot with all subjects together would be more informative; something like a 2D image of weights for all features selected across all subjects, with subjects on the y axis and features on the x axis, for example? This would

better show the commonalities across subjects, and would be a nicer summary for the reader.

- A. Thank you for this suggestion. We have revised Supplementary Appendices A and B to include carpet plots that show the distributions of weights associated with selected PLS models. We have also added a reference to these in the main manuscript in Section III.A (green colored text).

- 4. Thanks for adding the code and features data. This should definitely be uploaded on a public website and linked properly in the main text ; this constitute a very valuable addition. Some minor comments on the python code ; (a) the jupyter notebook with the univariate feature selection contains python imports which are not used. (b) It would be good to have a few more comments explaining what is done in the cells (c) as well as a very basic visualisation of the results, by plotting the p values (or the negative log p values) with the ROI labels, of such kind of visualization.

- A. We have streamlined the Jupyter notebook and added comments and visualizations for p-values. We will provide this code and data as part of the supplementary material that will be available publicly upon acceptance of this manuscript.

Reviewer 2

- 1. The revision of this paper is much improved. The re-organization and additional explanation make it easier to read and understand. The authors have addressed all comments related to the section on the PAI. I have no further comments and recommend this version of the manuscript for publication.

We thank the reviewer for their evaluation and feedback on our work.

Reviewer 4

- 1. The authors have done a reasonable job answering my previous questions. I have one question/comment that still remains. It would be interesting to discuss the authors graph signal processing which relies on having DTI data as well as fMRI data in comparison to newer approaches that do not rely on DTI data. See for example:

Gao S, Xia X, Scheinost D, Mishne G. Smooth graph learning for functional connectivity estimation. *Neuroimage*. 2021 Oct 1;239:118289. doi: 10.1016/j.neuroimage.2021.118289. Epub 2021 Jun 23. PMID: 34171497.

A. Thank you for this suggestion. We have added a reference to the suggested study and discussed it in the introduction (red colored text on page 4).

REVIEWERS' COMMENTS:

Reviewer #1 (Remarks to the Author):

Thanks a lot for the effort undertaken in improving this paper. The authors have addressed all remaining issues in the paper.

Reviewer #4 (Remarks to the Author):

The authors have addressed my comments.